# TNF stimulation primarily modulates transcriptional burst size of NF-κB-regulated genes

Victor L Bass[1],[†] (ID), Victor C Wong[1],[‡] (ID), M Elise Bullock[2] (ID), Suzanne Gaudet[3],[4],[#] &
Kathryn Miller-Jensen[1],[2],[*] (ID)

## Abstract

Cell-to-cell heterogeneity is a feature of the tumor necrosis factor (TNF)-stimulated inflammatory response mediated by the transcription factor NF-κB, motivating an exploration of the underlying sources of this noise. Here, we combined single-transcript measurements with computational models to study transcriptional noise at six NF-κB-regulated inflammatory genes. In the basal state, NF-κB-target genes displayed an inverse correlation between mean and noise characteristic of transcriptional bursting. By analyzing transcript distributions with a bursting model, we found that TNF primarily activated transcription by increasing burst size while maintaining burst frequency for gene promoters with relatively high basal histone 3 acetylation (AcH3) that marks open chromatin environments. For promoters with lower basal AcH3 or when AcH3 was decreased with a small molecule drug, the contribution of burst frequency to TNF activation increased. Finally, we used a mathematical model to show that TNF positive feedback amplified gene expression noise resulting from burst size–mediated transcription, leading to a subset of cells with high TNF protein expression. Our results reveal potential sources of noise underlying intercellular heterogeneity in the TNF-mediated inflammatory response.

**Keywords** inflammation; NF-κB; TNF; transcriptional bursting
**Subject Categories** Immunology; Signal Transduction
**Mol Syst Biol. (2021) 17: e10127**

## Introduction

Tumor necrosis factor (TNF) activates pro-inflammatory and stress response signaling in many cell types (Aggarwal, 2003). The TNF inflammatory response is mediated by the transcription factor NF-κB, which regulates the expression of hundreds of genes. These genes include inflammatory cytokines that can propagate an immune response via paracrine signaling, as well as negative regulators of NF-κB (Pahl, 1999; Hoffmann *et al*, 2002; Smale, 2011). Dysregulation of the TNF-stimulated NF-κB response contributes to inflammatory disease states (Lewis & Pollard, 2006; Schottenfeld & Beebe-Dimmer, 2006), and thus, NF-κB-induced transcription is tightly regulated in cell populations. However, it has been widely observed that TNF stimulates significant cell-to-cell heterogeneity in NF-κB signaling and in the transcription of its inflammatory gene targets (Tay *et al*, 2010; Cheong *et al*, 2011; Lee *et al*, 2014; Zhang *et al*, 2017; Wong *et al*, 2019). Although cell-to-cell heterogeneity in NF-κB signaling has been widely explored, additional sources of noise underlying transcription are not well understood. Understanding these sources of noise may enhance our ability to modulate the inflammatory response in clinically relevant ways.

One major source of single-cell gene expression noise is the fluctuation of promoters between transcriptionally active and inactive states, a process termed transcriptional bursting (Raj *et al*, 2006; Singh *et al*, 2010; Skupsky *et al*, 2010; Suter *et al*, 2011; Dar *et al*, 2012; Halpern *et al*, 2015b). Though gene expression noise can be buffered by various mechanisms (Halpern *et al*, 2015a; Padovan-Merhar *et al*, 2015; Stoeger *et al*, 2016), in some cases, it is amplified by regulatory networks to drive diverse cellular behaviors (Weinberger *et al*, 2005; Acar *et al*, 2008; Chang *et al*, 2008; Shalek *et al*, 2014). Several molecular mechanisms have been associated with transcriptional bursting including nucleosome positioning (Raser & O'Shea, 2004; Dey *et al*, 2015), chromatin modifications (Suter *et al*, 2011; Chen *et al*, 2019), transcription factor activity (Senecal *et al*, 2014; Li *et al*, 2018), and RNA polymerase (RNAPII) pause regulation (Wong *et al*, 2018; Bartman *et al*, 2019).

Although transcriptional bursting has not been extensively studied at endogenous NF-κB target genes, it has been well characterized for the HIV long terminal repeat (LTR) promoter, which is regulated

1 Department of Molecular, Cellular, and Developmental Biology, Yale University, New Haven, CT, USA
2 Department of Biomedical Engineering, Yale University, New Haven, CT, USA
3 Department of Cancer Biology and Center for Cancer Systems Biology, Dana-Farber Cancer Institute, Boston, MA, USA
4 Department of Genetics, Harvard Medical School, Boston, MA, USA
*Corresponding author (lead contact). Tel: +1 203 432 4265; E-mail: kathryn.miller-jensen@yale.edu
†Present address: Neuro-Immune Regulome Unit, National Eye Institute, National Institutes of Health, Bethesda, MD, USA
‡Present address: Janelia Research Campus, Howard Hughes Medical Institute, Ashburn, VA, USA
#Present address: Novartis Institute for BioMedical Research, Cambridge, MA, USA

by NF-κB. Transcriptional bursting at the HIV LTR has been shown to be influenced by chromatin environment both in the basal state (Singh *et al*, 2010; Dar *et al*, 2012; Dey *et al*, 2015), and after TNF stimulation (Dar *et al*, 2012; Wong *et al*, 2018). Specifically, it was shown that TNF could modulate either burst frequency (i.e., the rate of transition from an inactive to active state promoter state) or burst size (i.e., the number of transcripts produced per burst) of silent-but-inducible HIV LTR promoters, and that the bursting mechanism was influenced by the basal histone 3 acetylation state at the promoter (Wong *et al*, 2018). Endogenous NF-κB target promoters are found in basal chromatin environments that resemble those of latent-but-inducible HIV promoters (Ramirez-Carrozzi *et al*, 2009). Thus, we sought to determine whether molecular mechanisms regulating transcriptional bursting at inducible HIV LTRs are similar for endogenous NF-κB targets.

In this study, we analyzed changes in gene expression noise and transcriptional bursting at six endogenous NF-κB target promoters before and after TNF stimulation. We found that TNF stimulation increased mean transcription while maintaining noise for all but the most repressed NF-κB-target genes. We found that TNF stimulation primarily increased burst size while maintaining burst frequency, leading to highly skewed transcript distributions, especially for *Tnf* and *Il8*. Differences in basal histone acetylation at target promoters and RNA polymerase (RNAPII) pause regulation were associated with differences in the regulation of transcriptional bursting in response to TNF; and reducing basal histone acetylation at the *Tnf* promoter prior to stimulation caused TNF to shift from increasing burst size to increasing burst frequency. Finally, we used a mathematical model to explore how TNF positive feedback affects cell-to-cell heterogeneity in *Tnf* transcription. We found that transcription mediated via a burst size increase, as compared to a burst frequency increase, resulted in more heterogeneous cell populations when amplified by positive feedback, with a small subset of high TNF producers. Overall, we conclude that TNF primarily increases transcriptional burst size for endogenous NF-κB target promoters. Moreover, our results suggest that burst size–mediated transcription combined with positive feedback may contribute to the substantial cell-to-cell variability observed in the TNF-mediated inflammatory response.

# Results

## Single-molecule mRNA quantification reveals a conserved mean-noise relationship for TNF-NF-κB gene targets in the basal state

To characterize transcriptional noise in NF-κB targets induced by TNF, we analyzed six genes regulated by NF-κB. These genes have different roles in the TNF-induced inflammatory response. *Nfkbia* and *Tnfaip3* encode the intracellular proteins IκB-α and A20, respectively, which negatively regulate NF-κB p65 (Baeurerle & Baltimore, 1988; Heyniinck *et al*, 1999; Hoffmann *et al*, 2002). *Tnf*, *Il8*, *Il6*, and *Csf2* encode the secreted inflammatory cytokines TNF, IL-8, IL-6, and GM-CSF, respectively (Fig 1A). *Nfkbia*, *Tnfaip3*, *Tnf*, and *Il8* are classified as primary inflammatory genes because they are transcribed directly in response to stimulation in immune cells, while *Il6* and *Csf2* are classified as secondary genes because they require synthesis of additional protein regulators prior to transcription (Ramirez-Carrozzi *et al*, 2006; Hargreaves *et al*, 2009; Ramirez-Carrozzi *et al*, 2009).

To quantify transcription of these genes in cell populations, we treated the leukemic Jurkat T-cell line with TNF (20 ng/ml). Following TNF stimulation, *Nfkbia* and *Tnfaip3* exhibited the highest transcription, while *Tnf* and *Il8* were significantly lower, as measured in the population by RT–qPCR (Fig 1B). Increases in *Il6* and *Csf2* were not detectable in the population even 4 h after TNF stimulation. Notably, the differences in transcription were not due to differences in NF-κB p65 binding, because following TNF stimulation, NF-κB p65 promoter binding increased similarly across all promoters as measured by ChIP, including at the *Il6* and *Csf2* promoters (Fig 1C).

To quantify transcription in single cells, we performed single-molecule RNA fluorescence in situ hybridization (smFISH) in Jurkat T cells (Fig 1D and E and Appendix Fig S1) (Raj *et al*, 2008). We found very low levels of basal transcription, ranging from an average of 10 mRNAs per cell for *Nfkbia* to less than one mRNA on average per cell for *Il6* and *Csf2* (Fig 1F). We also observed significant cell-to-cell heterogeneity as measured by coefficient of variation (CV), with higher CV for the lower expression genes (Fig 1G). These genes are found in a range of basal chromatin environments, as quantified by the ratio of histone H3 acetylated at lysine 9 and 14

---

**Figure 1. Basal mean and transcriptional noise of NF-κB targets varies systematically with chromatin environment at the promoter.** ▶

A    NF-κB can recruit a variety of binding partners to target promoters, including the chromatin modifying enzyme p300, the elongation complex P-TEFb, and components of transcriptional machinery. NF-κB target genes with a variety of functions were chosen for this study.

B    Induction of NF-κB targets in Jurkat T cells in response to 20 ng/ml TNF treatment for 1, 2, and 4 h as measured by RT–qPCR. Target values were normalized to GAPDH and are reported as fold change relative to basal expression. Data are presented as mean ± standard deviation (SD) of three biological replicates.

C    Enrichment of RelA before and 30 and 60 min after treatment with 20 ng/ml TNF as measured by ChIP-qPCR and shown as % input (non-IP control). Data are presented as mean ± SD of three biological replicates.

D    Maximum intensity projections of smFISH fluorescence microscopy z-stacks of basal Jurkat T cells stained for the indicated genes. *Nfkbia*, *Tnfaip3*, *Tnf*, and *Il6* were labeled with Quasar 670, and *Il8* and *Csf2* were labeled with fluorescein. All images were filtered as described in Materials and Methods. Brightness and contrast were enhanced for visualization. Scale bars: 10 μm.

E    Histograms of transcripts per cell for target genes (blue) overlaid with probability density plots (red) generated from smFISH data. Cells were combined from three replicates (*Nfkbia*, *Tnfaip3*, *Tnf*, and *Il6*) or one replicate (*Il8*, *Csf2*).

F, G    Bar graphs of mean (F) and CV (G) of smFISH distributions for the indicated genes. Error bars indicate bootstrapped 95% confidence intervals (CIs) for the samples in (E). Significant differences indicated by non-overlapping CIs.

H    Ratio of enrichment of total histone H3 to acetylated H3 (AcH3) in Jurkat T cells at the indicated target promoters quantified by ChIP-qPCR. Data are presented as mean of % input (non-IP control) ± SD of three biological replicates.

I    Graph of $\log_{10}$(mean) vs $\log_{10}(CV^2)$ of basal mRNA distributions measured in Jurkat T cells (black), HeLa cells (red), or murine bone marrow–derived macrophages (green) for endogenous genes and four latent HIV LTR integrations. Gray shading indicates 95% CI of the linear regression for the basal trend line. Poisson trend line indicated by dashed line. HeLa data from Lee *et al* (2014) and HIV LTR data from Wong *et al* (2018).

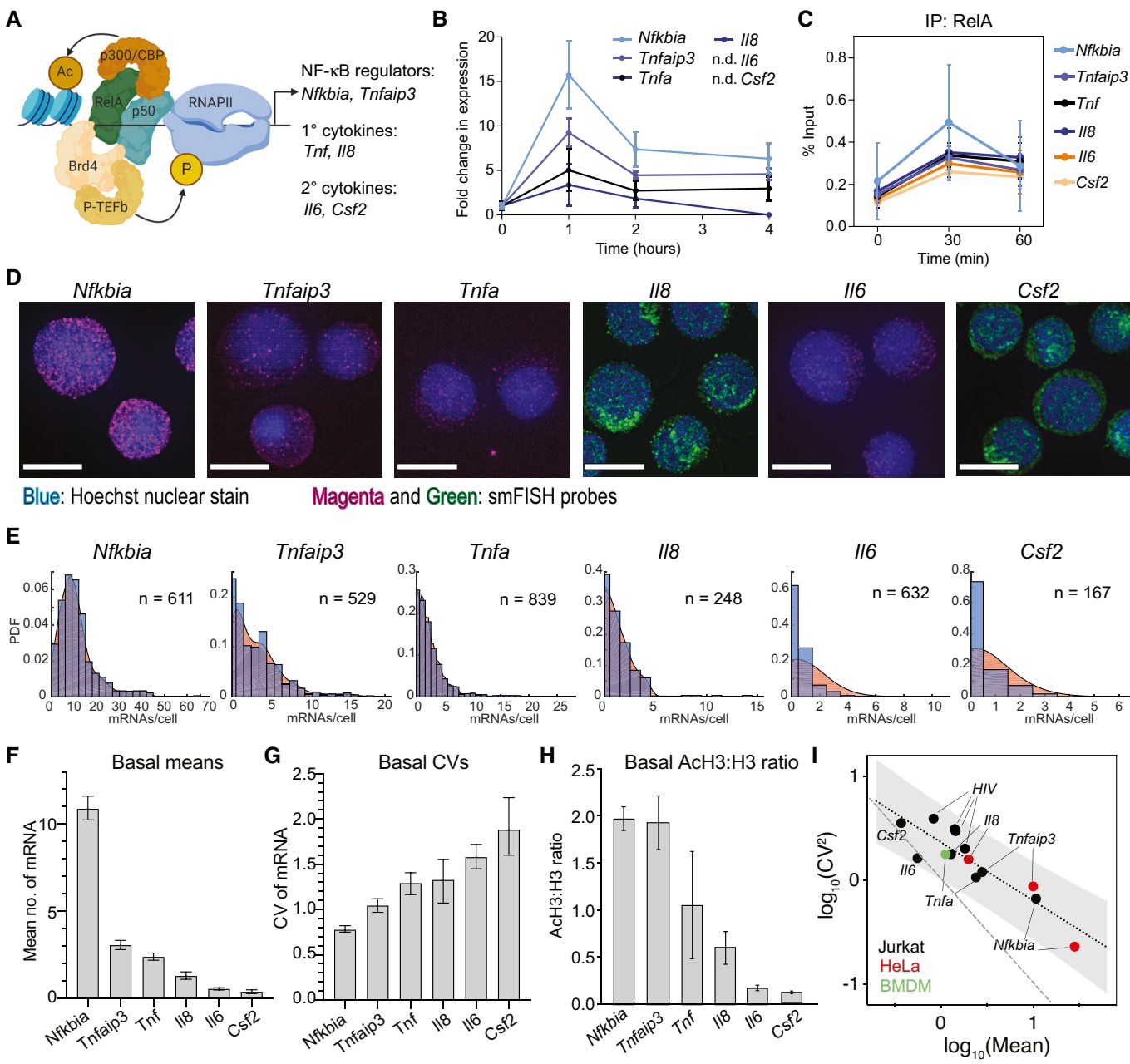

**Figure 1.**

(AcH3) to total histone H3 levels (AcH3:H3) at their promoters measured by chromatin immunoprecipitation (ChIP). *Nfkbia* and *Tnfaip3* had the highest ratio of AcH3:H3, indicating a more open chromatin environment, while *Il6* and *Csf2* had much lower ratios, indicating a more closed chromatin state (Fig 1H). We note that average basal mRNA levels increased monotonically with AcH3:H3 ratio, while CV decreased. Thus, the influence of chromatin state is apparent in the mean and variability of basal mRNA levels prior to TNF stimulation.

There is evidence for global constraints on transcriptional noise in mammalian cells (Sanchez & Golding, 2013), and our observation of systematic changes in mean and noise across NF-κB targets in

different chromatin environments is consistent with this hypothesis. To explore this further, we plotted the $\log_{10}$(mean) vs the $\log_{10}(CV^2)$ for the basal mRNA measurements of these six NF-κB-regulated genes in Jurkat cells (Fig 1I). Interestingly, when we plotted $\log_{10}$(mean) vs $\log_{10}(CV^2)$ for smFISH measurements of a subset of the same targets in HeLa cells (Lee *et al*, 2014) or in murine bone marrow-derived macrophages, we found that these measurements fell along the same line. Furthermore, basal mRNA measurements for exogenous HIV LTR promoters measured in Jurkat T cells and exhibiting similar basal chromatin states (Wong *et al*, 2018) also fell along the same trend line ($R^2 = 0.79$ for all points). Noise decreased as mean increased along this trend line, but the slope of this inverse

relationship was less steep than what would be expected from a Poisson distribution (Fig 1I), suggesting stochastic basal transcription rather than continuous transcription for inducible NF-κB targets (Singh et al, 2010; Skupsky et al, 2010; Dar et al, 2016). Altogether we conclude that there is a non-Poissonian relationship between basal transcriptional mean and noise that is conserved across NF-κB targets in multiple cell types.

## TNF stimulation differentially regulates transcriptional noise at NF-κB target genes

Genes in such diverse basal chromatin environments likely require recruitment of different factors by NF-κB to effectively activate transcription, which may lead to systematic differences in single-cell transcription distributions following stimulation (Neuert et al, 2013; Senecal et al, 2014). To analyze how transcriptional noise is altered by TNF-induced activation of NF-κB, we again quantified mRNA using smFISH (Fig 2A and Appendix Fig S1). For Nfkbia, Tnfaip3, Tnf, and Il8, we measured mRNA counts at 1- and 2-h post-TNF treatment to capture the peak and reduction in expression (Fig 2B). For Il6 and Csf2, we measured mRNA counts at 2- and 4-h post-TNF treatment when transcription was still rising. Notably, we were able to measure a significant increase in mRNA levels for Il6 and Csf2 by smFISH, even though increases in transcription were not detectable by population-level RT–qPCR (Fig 2B vs Fig 1B).

Although TNF treatment increased mean mRNA counts for all targets, the change in transcriptional noise varied by gene, as observed from the single-cell mRNA distributions (Fig 2C). After TNF treatment, Nfkbia, Tnfaip3, Tnf, and Il8 were expressed in most cells but at different levels, and all four targets exhibited long-tailed distributions, with a few cells expressing mRNA counts much higher than the mean. In contrast, Il6 and Csf2 were expressed at much lower levels with more non-expressing cells and exhibited less skewed distributions (Fig 2C). These differences in mRNA distributions across targets were apparent when observing the dynamic trends in CV. For Nfkbia, Tnfaip3, Tnf, and Il8, the CV of mRNA counts remained relatively constant from 0 to 2 h, while the CV of mRNA counts for Il6 and Csf2 decreased from 0 to 4 h (Fig EV1A).

Recent literature suggests that some transcript heterogeneity may be due to extrinsic factors including cell size and cell cycle state and may be buffered by nuclear export (Battich et al, 2015; Halpern et al, 2015a; Padovan-Merhar et al, 2015; Stoeger et al, 2016). We compared nuclear and cytoplasmic noise before and after TNF stimulation and observed a minor attenuation of noise that may be attributed to transcription occurring more quickly than nuclear

export of mRNA immediately after stimulation (Hansen et al, 2018) (Fig EV1B). In general, CVs of cell area and nuclear area, which we used as a proxy for cell cycle (Padovan-Merhar et al, 2015; Chu et al, 2017), were less than transcript CVs; and normalizing Tnfaip3, Tnf, and Il6 transcript counts by cell or nuclear area did not significantly reduce noise (Fig EV1C). We also looked for evidence of shared sources of noise from upstream signaling regulators in the TNF-NF-κB pathway by measuring Nfkbia and Tnf in the same cells using multiplexed smFISH. We found only a moderate correlation between these two targets ($r = 0.34$, $P < 0.001$) that decreased after 2 h of TNF stimulation ($r = 0.14$, $P = 0.09$; Fig EV1D). Our observed lack of correlation with cell size and the relatively low correlation between Nfkbia and Tnf was different from what was observed previously for the same targets following LPS stimulation in macrophages (Bagnall et al, 2018). This is likely attributable to differences in cell type and stimulus, as well as the more than 10-fold lower gene expression observed for our targets. Overall, our results suggest that shared sources of cellular variation—including nuclear export, cell size, cell cycle, and shared upstream signaling regulators—do not fully account for the gene-specific noise observed in our experiments.

To visualize how TNF-NF-κB-mediated transcription changed the global mean-noise relationship seen in the basal state, we plotted $\log_{10}(\text{mean})$ and $\log_{10}(CV^2)$ of mRNA counts before and after TNF treatment. For the NF-κB targets that increased mean without a significant reduction in noise (i.e., Nfkbia, Tnfaip3, Tnf, and Il8), we observed that in some cases the points moved outside the basal trend line resulting in noise that further deviated from Poissonian behavior (Fig 2D). In contrast, TNF treatment for 2 and 4 h caused an increased mean with a concomitant decrease in noise in Il6 and Csf2 that was consistent with the basal trend line (Fig 2E). Overall, these trends suggest that NF-κB differentially regulates transcriptional noise at different target genes following TNF stimulation.

## TNF stimulation primarily modulates burst size of NF-κB targets

For many mammalian genes, transcription occurs in short bursts. Transcriptional bursting behavior can be effectively modeled with two promoter states, in which a promoter briefly switches from an "OFF" state to a transcript-producing "ON" state, before switching back to the "OFF" state (Fig 3A) (Raj et al, 2006; Singh et al, 2010; Skupsky et al, 2010; Suter et al, 2011; Dar et al, 2012; Halpern et al, 2015b). In this model, the transcriptional process is described by two main features: burst size, defined as the average number of mRNA produced per burst (i.e., gene activation event), and burst

**Figure 2.  TNF induces gene-specific changes in transcript distributions at NF-κB targets.**

A    Maximum intensity projections of smFISH fluorescence microscopy z-stacks of Jurkat T cells stained for the indicated genes after 1-h (Nfkbia, Tnfaip3, Tnf, Il8) or 4-h (Il6, Csf2) treatment with 20 ng/ml TNF. Nfkbia, Tnfaip3, Tnf, and Il6 were labeled with Quasar 670, and Il8 and Csf2 were labeled with fluorescein. All images were filtered as described in Materials and Methods. Brightness and contrast were enhanced for visualization. Scale bars: 10 μm.

B    Bar graphs of mean of smFISH distributions before and after TNF treatment for the indicated genes. Cells were combined from three replicates (Nfkbia 1 h; Tnfaip3 1 h; Tnf 1, 2 h; and Il6 2, 4 h) two replicates (Nfkbia 2 h) or one replicate (Tnfaip3 2 h; Il8 1, 2 h; Csf2 2, 4 h). Basal data are same as in Fig 1E. Error bars indicate bootstrapped 95% CIs for the samples in (C). Significant differences indicated by non-overlapping CIs.

C    Probability density plots of single-cell mRNA distributions from smFISH as described in (B) before and after treatment with 20 ng/ml TNF for the indicated time points.

D, E    Graph of $\log_{10}(\text{mean})$ vs $\log_{10}(CV^2)$ for endogenous gene targets that maintain $CV^2$ (D) or decrease $CV^2$ (E) after treatment with 20 ng/ml TNF in Jurkat T cells (black), HeLa cells (red), or murine bone marrow–derived macrophages (green). Gray shading indicates 95% CI of basal trend line. Poisson trend line indicated by dashed line. HeLa data from Lee et al (2014).

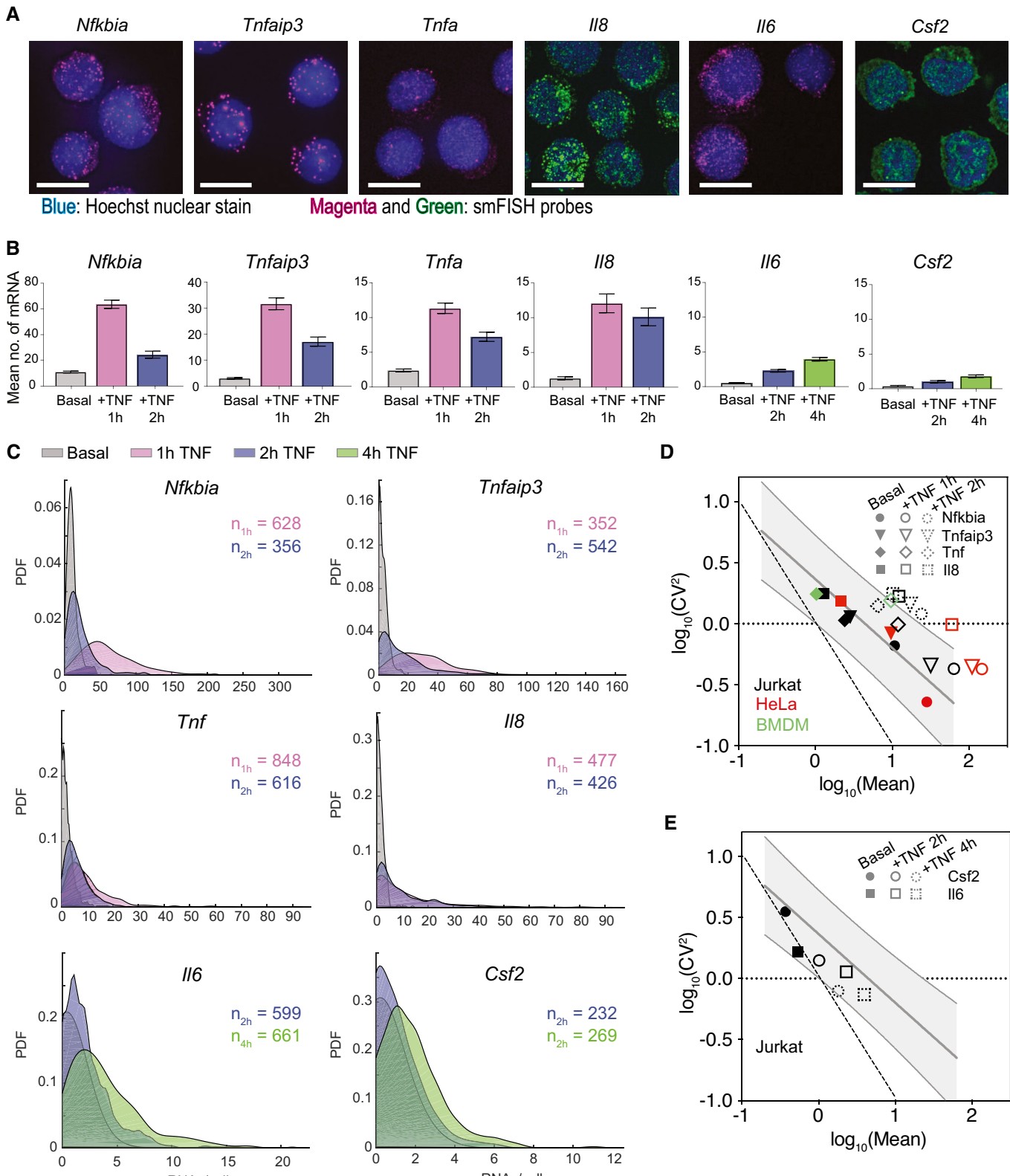

**Figure 2.**

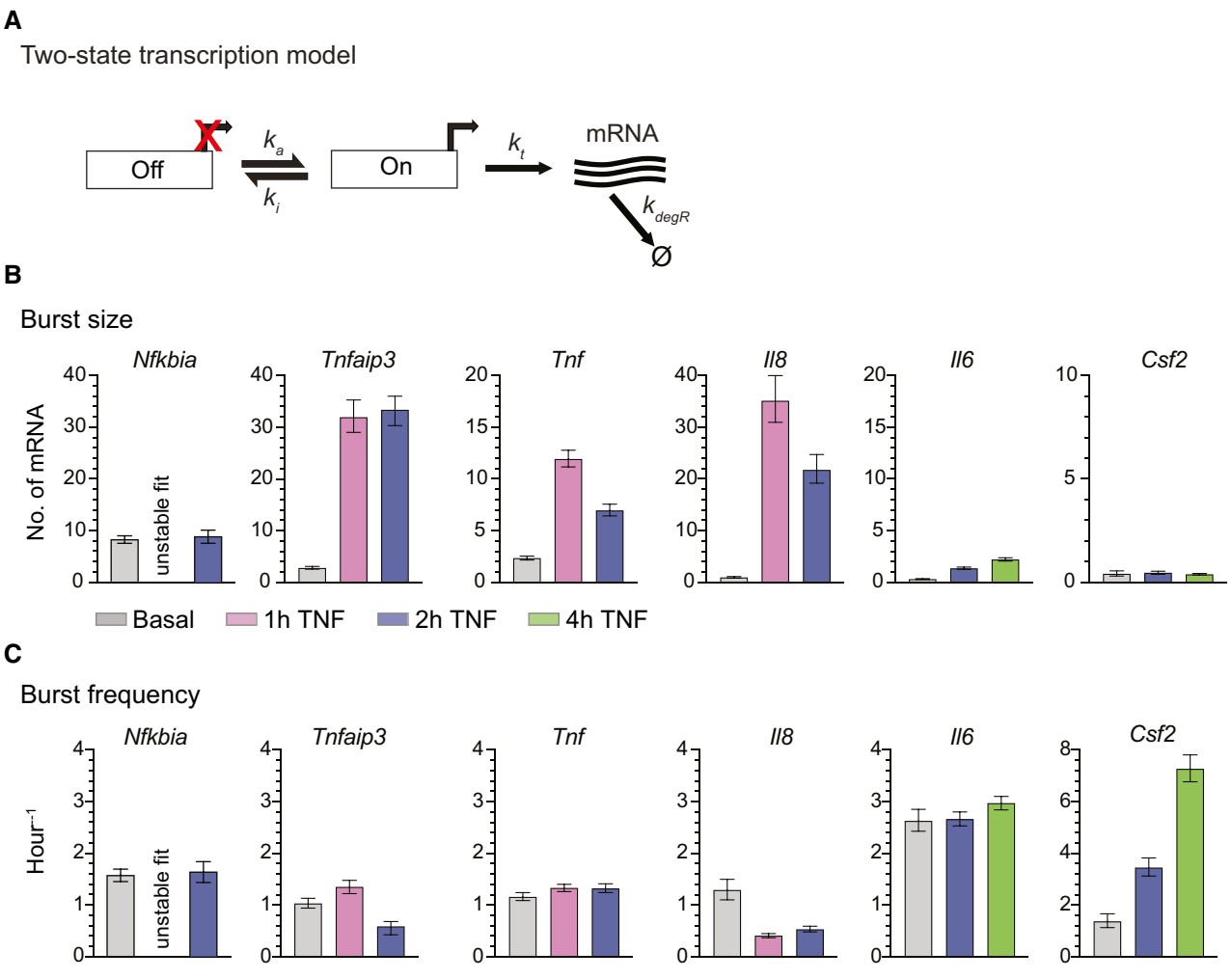

**Figure 3. Inferred fits from two-state promoter model show that TNF treatment increases transcriptional burst size at most targets.**

A   Schematic of a two-state promoter model for transcriptional bursting. Burst frequency ($k_a$) and burst size ($b = k_t/k_i$) were fit to combined transcript distributions measured by smFISH.

B, C   Burst size (B) and burst frequency (C) parameter fits from the two-state model in the basal state and after treatment with 20 ng/ml TNF for 1, 2, or 4 h. Error bars indicate bootstrapped 95% CIs. Significant differences indicated by non-overlapping CIs. The fit for *Nfkbia* (1-h TNF) was unstable and thus is not reported (see Materials and Methods). Sample sizes are displayed in Figs 1E and 2C.

frequency, defined as the frequency with which the bursts occur (Nicolas *et al*, 2018). The observed mean-variance and mean-noise trends of the basal and TNF-stimulated transcript distributions of the target genes show deviation from Poisson behavior that is consistent with transcriptional bursting (Fig EV2A) (Singh *et al*, 2010; Skupsky *et al*, 2010; Wong *et al*, 2018; Bagnall *et al*, 2020). Thus, we expected the transcriptional bursting model would provide insight into the observed differences in transcriptional noise across NF-κB target genes.

We first estimated burst size and burst frequency using the sample variance $\sigma^2$ and the mean µ of the mRNA distributions (Raj *et al*, 2006; Suter *et al*, 2011; Nicolas *et al*, 2018; Bagnall *et al*, 2020). These estimates of burst size $b_m = \sigma^2/\mu$ (i.e., the Fano factor) and burst frequency $f_m = \mu/(b_m - 1)$ based on the sample moments

are often used to describe deviation from Poisson distributions, for which $b_m = 1$ and $f_m = \infty$ (So *et al*, 2011; Nicolas *et al*, 2018; Bagnall *et al*, 2020). Analysis of burst size and burst frequency based on the moments of the smFISH count distributions revealed differences in how TNF affected *Nfkbia, Tnfaip3, Tnf* and *Il8* vs *Il6* and *Csf2*, with the former group exhibiting large increases in burst size (Fig EV2B) and the latter group exhibiting much smaller increases in burst size but with increasing values of burst frequency (Fig EV2C).

To further explore these differences, we fit our data to a two-state model of promoter activity (Raj *et al*, 2006; Dey *et al*, 2015) (Fig 3A). In this model, also known as the random telegraph model, transcription is described by four parameters: rate of transition to the active state, $k_a$; rate of transition to the inactive state, $k_i$; rate of

transcription in the active state, $k_t$; and mRNA degradation rate, $k_{deg}$. The probability density function (pdf) of this distribution can be solved theoretically and then burst frequency ($k_a$) and burst size (mean number of transcripts produced per active state burst, $b = k_t/k_i$) can be inferred by finding the optimum fit between the experimental and theoretical pdfs using maximum likelihood estimation (MLE) (Raj *et al*, 2006; Dey *et al*, 2015; Wong *et al*, 2018). To perform MLE, we fixed mRNA decay rate ($k_{deg}$) to experimentally measured values when possible. Transcription of *Il6* and *Csf2* was too low to be measured accurately, and so we used the average decay rate measured for the other four targets, which displayed similar transcript stability ($t_{1/2} \approx 40$ min) and is in line with previously reported values (Paschoud *et al*, 2006) (Appendix Fig S2A). We then fit burst size (the ratio of $k_t/k_i$) and burst frequency ($k_a$; see Materials and Methods).

When we fit our single-cell transcript distributions before and after TNF treatment to the two-state model's pdf, we found that the model fit all basal distributions and most TNF-stimulated distributions (Fig EV3). The one exception was the 1-h TNF-stimulated *Nfkbia* distribution, for which the theoretical pdf two-state model solution produced unstable fits (see Materials and Methods). These samples, which exhibit the largest transient increase in transcription induced by TNF, may not be well described by the random telegraph model. Model fits indicate that in the basal state, most genes share a low basal burst frequency of ~ 1 transition per hour and a burst size of only a few transcripts (Fig 3B and C), similar to $b_m$ and $f_m$ estimated by the moments (Fig EV2B and C). TNF treatment drives large increases in burst size with minimal changes in burst frequency for *Tnfaip3*, *Tnf*, and *Il8*. In contrast, TNF causes a small increase in both burst size and frequency for *Il6*, and a large increase in burst frequency with no change in burst size for *Csf2* (Fig 3B). Comparing these observations with AcH3:H3 ratios in the basal state (Fig 1H), we find that TNF stimulation primarily alters the burst size of promoters that exhibit high basal AcH3:H3 ratios. In contrast, for *Il6* and *Csf2*, which exhibit much lower AcH3:H3 ratios in the basal state, TNF stimulation only modestly increases burst size or, in the case of Csf2, increases burst frequency.

## TNF-mediated increases in burst size are associated with higher promoter levels of AcH3 and RNAPII pausing

Activation of a range of transcription factors (TFs) has been associated with changes in burst frequency for many genes (Li *et al*, 2018; Chen *et al*, 2019; Friedrich *et al*, 2019), while TF-mediated changes in burst size are less widely reported. However, our results are consistent with our previously reported observations at HIV LTRs integrated in different chromatin environments (Wong *et al*, 2018) and suggest that mechanisms of transcriptional bursting are affected by the chromatin state at the promoter. To search for potential differences in molecular events linked to changes in burst size after TNF treatment, we measured chromatin features and binding of transcriptional machinery at our target promoters using ChIP. Changes in transcriptional burst frequency have been linked to histone acetylation (Nicolas *et al*, 2018; Chen *et al*, 2019), and we previously showed that TNF-NF-κB-mediated increases in burst size at the HIV LTR were associated with regulation of RNAPII activity (Wong *et al*, 2018). Therefore, we focused on measuring histone H3 acetylation and markers of RNAPII regulation.

We first examined histone H3 acetylation at the target promoters by measuring total and acetylated H3. After TNF treatment, the secondary cytokines *Il6* and *Csf2* exhibited large decreases in total H3, while *Tnf* exhibited smaller decreases, so that by 4 h after TNF treatment, all targets had similar H3 levels (Fig 4A). In contrast, *Nfkbia*, *Tnfaip3*, and *Il8* significantly increased AcH3 following TNF treatment but did not exhibit significant changes in total H3. Chromatin remodeling is a molecular step that likely occurs before RNAPII regulation (Bartman *et al*, 2019), and so basal differences in histone acetylation might underlie the differential changes we see in bursting. Related to this, we note that *Il6* and *Csf2*, which exhibited the largest decreases in H3, exhibited the smallest increases in transcription overall and this increase was associated with higher burst frequencies.

We also measured total RNAPII, serine-5 phosphorylated RNAPII (ser5-p), serine-2 phosphorylated RNAPII (ser2-p), and negative elongation factor (NELF) before and at 2 and 4 h after TNF treatment (Fig 4A). We found that *Il6* and *Csf2* accumulated less total RNAPII than *Nfkbia*, *Tnfaip3*, *Tnf*, and *Il8*, which is consistent with the lower expression levels of these genes after TNF treatment. The disparity in RNAPII enrichment was lessened when looking at ser2-p RNAPII (associated with elongation) and heightened when looking at ser5-p RNAPII (associated with initiation). Enrichment of NELF, which inhibits elongation, coupled with enrichment of ser5-p RNAPII, is indicative of paused RNAPII at *Tnfaip3*, *Tnf*, and *Il8*, in contrast to the *Il6* and *Csf2* promoters. Taken together, the RNAPII ChIP shows that the *Tnfaip3*, *Tnf*, and *Il8* promoters, which increase burst size after TNF treatment, accumulate more paused RNAPII than the *Il6* and *Csf2* promoters in response to TNF.

Clustering our ChIP data, we found clear separation between *Il6* and *Csf2* and the more highly activated targets that show significant increases in burst size (Fig 4B). Within the non-burst frequency increasing genes, *Nfkbia* separates from all other genes due to its increased accumulation of RNAPII, and the primary cytokines *Tnf* and *Il8* separate out from *Tnfaip3*. The clustering supports the idea that differences in molecular events occur at promoters of genes that have increased burst frequency (*Csf2*, *Il6*) vs burst size (*Tnfaip3*, *Tnf*, *Il8*, and *Nfkbia*).

## Small molecule inhibitors of histone acetylation and RNAPII pause release alter TNF-mediated transcriptional bursting

Our ChIP data suggested an association between basal H3 acetylation (AcH3) at a target promoter and transcriptional bursting in response to TNF. Specifically, we observed that as basal AcH3 at the promoter increased, there was a shift toward a TNF-mediated increase in burst size (Fig 4B). This is consistent with previous work demonstrating that burst initiation (associated with burst frequency) precedes polymerase recruitment (associated with burst size; Bartman *et al*, 2019). Thus, we hypothesized that if we reduced basal AcH3 at target promoters prior to TNF stimulation, targets that previously exhibited large TNF-mediated increases in burst size (e.g., *Tnfaip3* and *Tnf*) would instead exhibit TNF-mediated increases in burst frequency and a reduced burst size increase.

To test this hypothesis, we perturbed basal AcH3 at *Tnfaip3* and *Tnf* promoters by pretreating Jurkat cells with the histone acetyltransferase (HAT) inhibitor A-485, a specific inhibitor of the HATs p300/CBP that are recruited by NF-κB (Fig 5A) (Gerritsen *et al*,

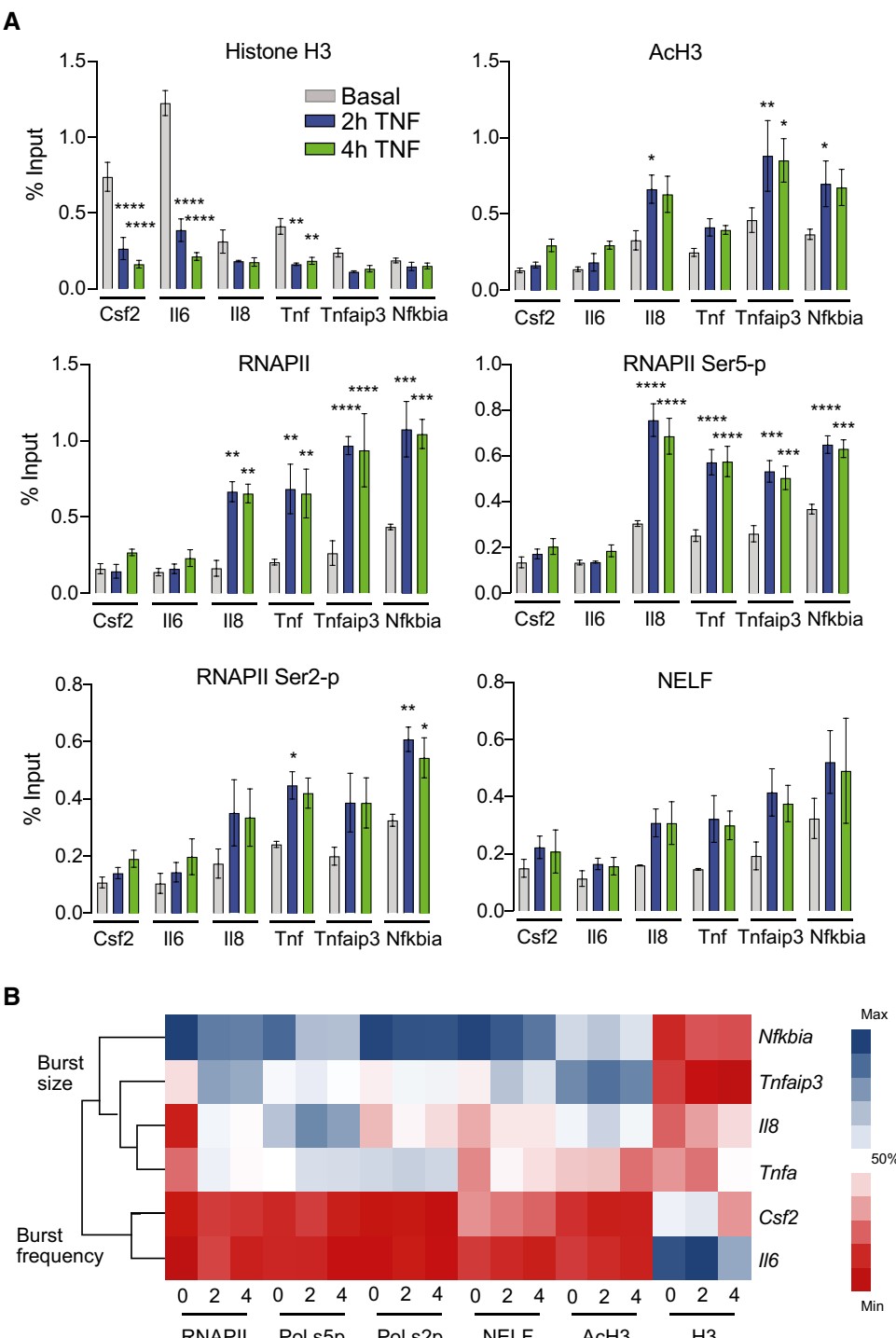

**Figure 4. RNAPII pausing is associated with increases in transcriptional bursting upon TNF treatment.**

A Enrichment of histone H3, AcH3, total RNPII, ser5-p RNPII, ser2-p RNPII, and NELF-E in the basal state (0 h) and after treatment (2 and 4 h) with 20 ng/ml TNF quantified using ChIP and shown as % input (non-IP control). Data are presented as mean ± standard error of the mean (s.e.m.) of three biological replicates. Significance calculated by Dunnett's multiple comparison test (*$P < 0.05$, **$P < 0.01$, ***$P < 0.001$, ****$P < 0.0001$).

B Hierarchical clustering of ChIP data before and after TNF treatment separates promoters with TNF-mediated increases in burst frequency or burst size. For each protein target, color bar indicates the % maximum ChIP value measured across all genes and time points.

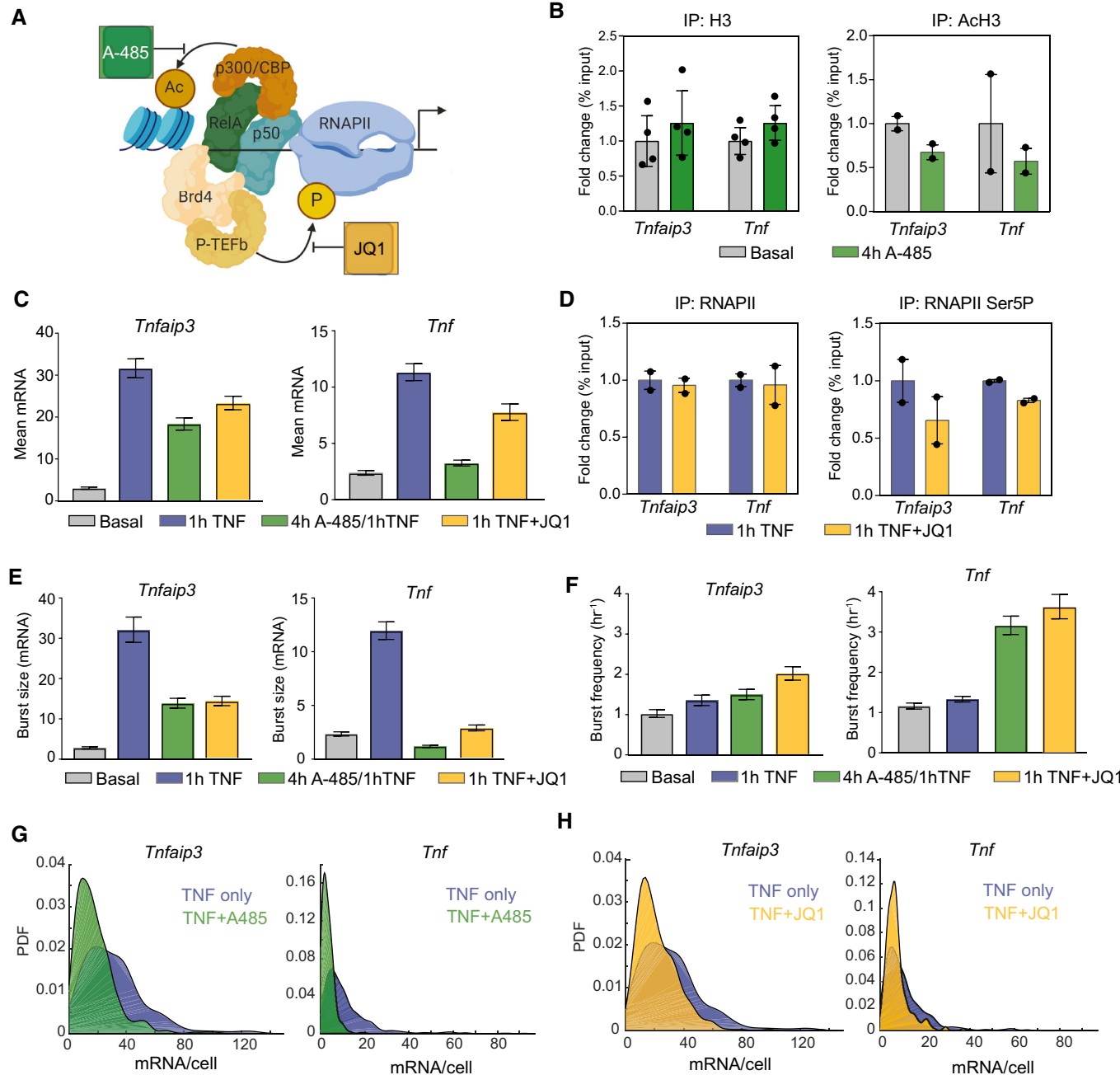

**Figure 5. Small molecule inhibitors of H3 acetylation and RNAPII pause release alter TNF-mediated changes in transcriptional bursting.**

A    Schematic of A-485 inhibition of the histone acetyl transferase p300/CBP, which is recruited by NF-κB, and of JQ1 inhibition of BET bromodomains, which recruit the positive transcription elongation factor b (P-TEFb).

B    Change in enrichment of histone H3 and AcH3 after treatment with 300 nM A-485 for 4 h measured by ChIP-qPCR and shown as % input (non-IP control) normalized to the uninhibited control for each gene. Data are presented as mean ± SD for two or four biological replicates.

C    Bar graphs of mean mRNA level for basal condition, 1-h TNF, 1-h TNF + 4 h pretreatment with 300 nM A-485, and 1-h TNF + 62.5 nM JQ1 cotreatment measured by smFISH for *Tnfaip3* (left) and *Tnf* (right). Cells were combined from two replicates (*Tnfaip3* A-485, JQ1; *Tnf* A-485) or one replicate (*Tnf* JQ1). Error bars indicated bootstrapped 95% CIs. Samples with non-overlapping CIs are significant.

D    Change in enrichment of total and Serine-5-phosphorylated RNAPII after 20 ng/ml TNF treatment for 1 h with 62.5 nM JQ1 measured by ChIP-qPCR and shown as % input (non-IP control) normalized to the uninhibited control for each gene. Data are presented as mean for two biological replicates.

E, F   Burst size (E) and burst frequency (F) parameter fits from the two-state model for the same conditions in (C) measured by smFISH for *Tnfaip3* (left) and *Tnf* (right) for the data described in (C). Error bars indicated bootstrapped 95% CIs. Samples with non-overlapping CIs are significant.

G, H   Probability density of mRNA distributions measured by smFISH for 20 ng/ml TNF for 1-h (blue) vs 1-h TNF + 4 h pretreatment with 300 nM A-485 (green) (G) and 1-h TNF (blue) vs 1-h TNF + 62.5 nM JQ1 cotreatment (yellow) (H) for *Tnfaip3* (left) and *Tnf* (right).

1997; Lasko *et al*, 2017). We found that pretreatment with A-485 for 4 h decreased AcH3 levels at the *Tnfaip3* and *Tnf* promoters, consistent with inhibition of HAT activity, but did not affect total H3 levels as measured by ChIP-qPCR (Fig 5B).

We then compared transcription 1 h after TNF stimulation with and without A-485 pretreatment. Overall, we found that A-485 pretreatment significantly reduced mean mRNA expression for both *Tnfaip3* and *Tnf* in response to TNF (Fig 5C, green). Consistent with our expectations, A-485 pretreatment reduced the moment burst size $b_m$ (as estimated by Fano factor) in response to TNF stimulation for *Tnf* and *Tnfaip3*, although the effect was more pronounced for *Tnf* (Fig EV4A). A-485 pretreatment increased the TNF-mediated change in the moment burst frequency $f_m$ for *Tnf* but not for *Tnfaip3* (Fig EV4B). Fitting mRNA distributions to the theoretical pdf of the two-state model further confirmed that A-485 pretreatment decreased burst size but did not affect burst frequency following TNF stimulation for *Tnfaip3*, while it decreased burst size and increased burst frequency for *Tnf* (Fig 5E and F, green). Overall, when AcH3 was reduced at the *Tnfaip3* promoter, its transcriptional bursting response resembled that of *Tnf* and *Il8* (no change in burst frequency combined with a smaller increase in burst size). Similarly, the reduction in AcH3 at the *Tnf* promoter resulted in transcriptional bursting that resembled *Il6* and *Csf2* (no change in burst size combined with an increased burst frequency). Thus, we conclude that decreasing basal AcH3 at target promoters shifts TNF-induced transcription from increasing burst size to increasing burst frequency.

Our ChIP data also suggested an association between RNAPII pausing at target promoters and TNF-mediated transcriptional bursting. Specifically, we observed that TNF-mediated increases in burst size were associated with increased RNAPII promoter-proximal pausing as measured by the accumulation of ser5-p RNAPII (Fig 4 B), perhaps because release of paused promoters produces a larger burst of transcription. To perturb RNAPII pause regulation, we treated Jurkat cells with JQ1, an inhibitor of the BET family of bromodomain proteins, including BRD4, which recruits the positive transcription elongation factor b (p-TEFb) that stimulates pause release (Fig 5A) (Huang *et al*, 2008; Hargreaves *et al*, 2009; Filippakopoulos *et al*, 2010). Previous work showed that JQ1 can inhibit multiple facets of gene regulation, including polymerase pause release and enhancer activity (Belkina & Denis, 2012; Shi & Vakoc, 2014; Stonestrom *et al*, 2016). When bursting was previously analyzed following treatment with JQ1, it was found to decrease both the rate of burst initiation and polymerase pause release, but it did not appear to change the rate of RNAPII recruitment (Bartman *et al*, 2019). Thus, we expected to observe a reduction in burst frequency and also burst size upon TNF stimulation in combination with JQ1.

We found that JQ1 treatment in combination with TNF stimulation decreased ser5-p RNAPII accumulation at the *Tnfaip3* and *Tnf* promoters as measured by ChIP-qPCR, but did not affect total RNAPII (Fig 5D). As expected, JQ1 concomitantly decreased TNF-stimulated mean expression of both *Tnfaip3* and *Tnf* (Fig 5C, yellow). JQ1 reduced $b_m$ for both *Tnfaip3* and *Tnf* (Fig EV4A), while $f_m$ was unchanged for *Tnfaip3* and increased for *Tnf* (Fig EV4B). Fitting mRNA distributions to the theoretical pdf of the two-state model confirmed that JQ1 reduced TNF-induced burst size increases for both *Tnfaip3* and *Tnf* (Fig 5E and F, yellow). However,

model fits further confirmed an increase in burst frequency for both genes, in contrast to expectations. Our data appear to confirm the multifactorial activity of JQ1, but are hard to interpret biologically.

Somewhat surprisingly, A-485 pretreatment and JQ1 cotreatment similarly affected TNF-mediated transcriptional activation for *Tnfaip3* and *Tnf*. However, when directly comparing single-cell mRNA distributions, we noted that the overall decrease in TNF-stimulated expression caused by A-485 pretreatment was marked by a greater increase in *Tnf* non-expressing cells than we observed for JQ1, consistent with a molecular mechanism in which histone acetylation at the promoter precedes RNAPII recruitment and pausing (Figs 5G and H and EV4C). Overall, we conclude that that basal histone AcH3 levels at NF-κB target promoters affect how TNF treatment alters transcriptional bursting, but more specific perturbations will be required to determine how TNF-stimulated accumulation of paused RNAPII at target promoters is linked to transcriptional burst size.

## Mathematical modeling predicts that TNF positive feedback can amplify distributions produced by transcriptional bursting to create more heterogeneous cell populations

TNF modulates transcriptional burst size at some promoters and burst frequency at others, producing more or less skewed mRNA distributions across a cell population, respectively. We recently showed that for latent-but-inducible integrations of the human immunodeficiency virus (HIV) in Jurkat T cells, the skewed HIV mRNA distributions produced by TNF activation of transcription via burst size resulted in viral activation when amplified by HIV-mediated positive feedback, while transcription via burst frequency did not (Wong *et al*, 2018). TNF positively regulates its own production analogous to HIV, and our results show that TNF does this by increasing transcriptional burst size. Therefore, we sought to explore whether modulation of burst size combined with positive feedback could further amplify cell-to-cell heterogeneity of TNF production.

To determine whether extracellular signaling amplifies *Tnf* transcription, we stimulated Jurkat cells with TNF in the presence of brefeldin A (BFA), which inhibits protein transport from the endoplasmic reticulum to the Golgi and thus blocks secretion. We found that BFA modestly reduced transcription at 2 h following TNF stimulation and also reduced the inferred burst size, while also increasing the inferred burst frequency (Fig EV5A). To determine whether this small difference in transcription led to measurable differences at the protein level, we measured intracellular TNF protein by flow cytometry following TNF stimulation in the presence of BFA for up to 8 h (to prevent all paracrine signaling) and for only the final 4 h of an 8-h TNF stimulation (to allow the first 4 h of paracrine signaling to occur). The fraction of responding cells was small, consistent with the low mRNA measurements, but a significant increase in intracellular TNF over control was seen after TNF stimulation (Fig EV5B). Importantly, we saw an increase in % TNF$^+$ cells at 8 h when BFA was withheld for 4 h to allow paracrine signaling to occur (Fig EV5B; although increase not statistically significant). Taken together, these data support a role for positive feedback in amplifying the response.

We built a mathematical model of a two-state *Tnf* promoter responding to an initial TNF stimulus and further amplified by positive feedback (Fig 6A). We modeled the addition of exogenous TNF

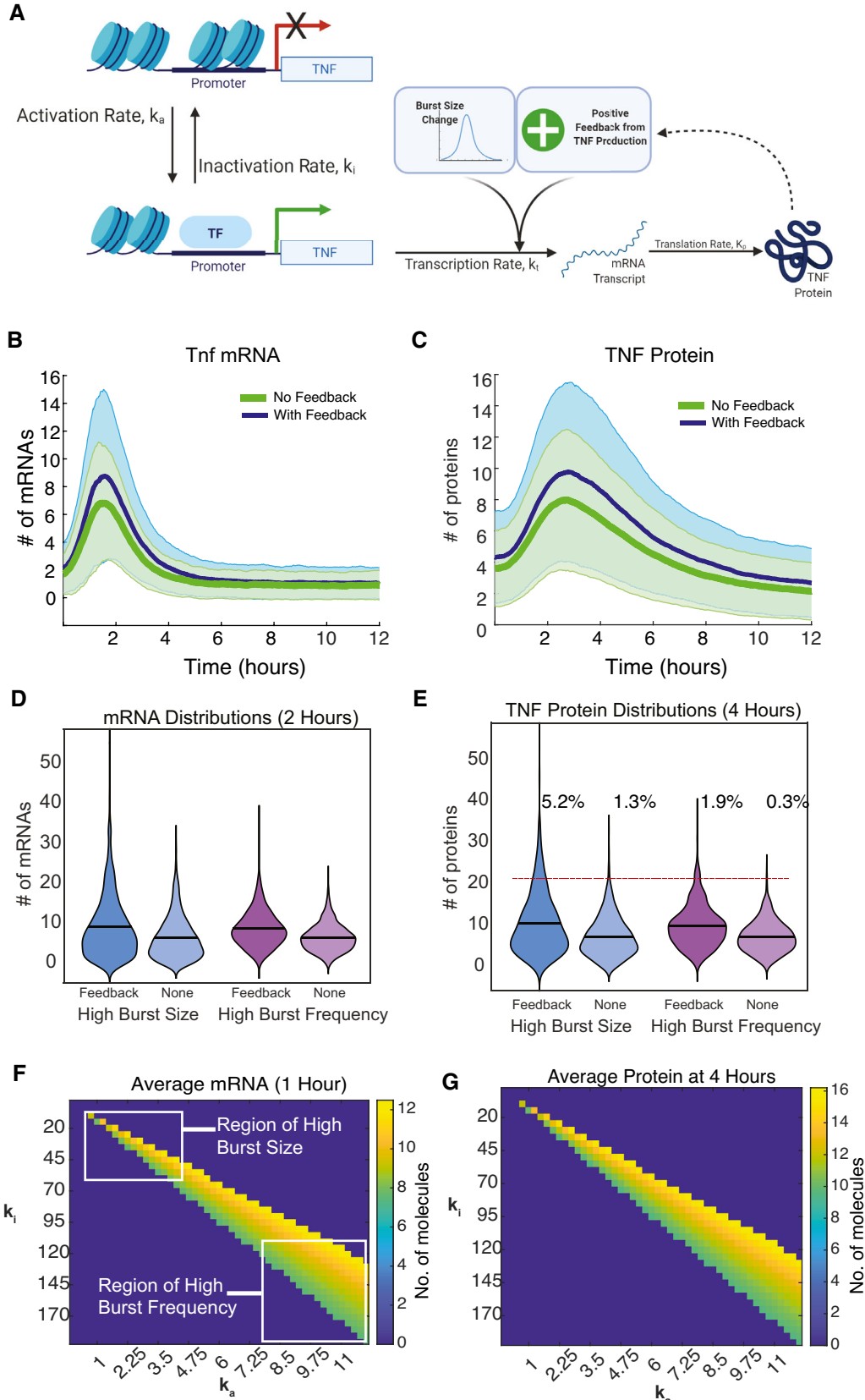

**Figure 6.**

**Figure 6.    Mathematical modeling predicts differences in cell-to-cell heterogeneity resulting from TNF positive feedback on burst size- vs burst frequency-mediated transcript activation.**

A        Schematic of the two-state model of transcription coupled to translation of a protein (TNF) that positively feeds back on its own transcription rate.
B, C    Model simulation of cell-population averages of *Tnf* mRNA (B) and TNF protein (C) vs time with and without positive feedback. Data are presented as mean (dark line) and SD (shaded region) of 1,000 simulated cells.
D, E    Violin plots of simulated single-cell mRNA (D) and protein (E) numbers with and without positive feedback for either an increase in burst size (blue) or burst frequency (purple) in response to exogenous TNF treatment. Data are presented as number of molecules in 1,000 simulated cells. Red line indicates threshold of 20 protein molecules designating activated cells.
F, G    Cell-population averages from stochastic simulations with positive feedback of *Tnf* mRNA at 1 h (F) and TNF protein at 4 h (G) after TNF treatment across a parameter space with increasing burst frequency ($k_a$) and decreasing burst size ($k_i$) chosen to produce levels of *Tnf* mRNA similar to experimental observations.

as a time-dependent change in $k_t$, the mRNA production rate. We fit this $k_t$ function empirically to match the TNF-induced change in burst size inferred from our smFISH distributions over time, including the effect of BFA at 2 h (Fig EV5A and C). We then explored a range of TNF positive feedback parameters (Fig EV5D) and identified values that qualitatively reproduced the dynamic changes in cell-population averages and distributions observed in our population-level RT–qPCR measurements of *Tnf* transcription in the presence of feedback (Figs 6B and EV5E). When we examined the results of our simulations, we found that positive feedback increased the level and variability of TNF protein (Fig 6C). By 4 h, our simulations showed that TNF positive feedback increased the small subset of high-producing TNF cells (Fig 6D and E, blue), similar to what we observed experimentally (Fig EV5B).

We then sought to explore how positive feedback would amplify TNF-stimulated transcriptional increases of similar means but with different noise. To do this, we performed a parameter scan in the absence of positive feedback, in which we increased burst frequency (i.e., the activation rate $k_a$) and simultaneously decreased burst size (i.e., by increasing the inactivation rate $k_i$). By increasing burst frequency while simultaneously decreasing burst size, we were able to identify a region in which mean expression remains relatively constant but noise varies due to differences in burst behavior (Fig 6 F and G). Positive feedback still increased the small subset of high-producing TNF cells at 4 h for the "burst-frequency" parameter set; however, it was a smaller absolute increase as compared to the mRNA distribution with the "burst size" parameter set (Fig 6D and E, purple vs blue). In other words, the small population of high TNF-producing cells was more pronounced when transcription was increased via burst size vs burst frequency. The effect of positive feedback on increasing cell-to-cell heterogeneity was evident in the large increase in Fano factor that was greatest for the "high burst-size" parameters (Fig EV5F). Overall, our modeling indicates that positive feedback by TNF coupled with transcriptional increases driven by burst size modulation can produce highly skewed distributions of protein across cells. We speculate that these mechanisms could contribute to small subpopulations of cells with high functionality, such as high cytokine-producing cells that have been observed in response to activation of the NF-κB-mediated inflammatory response in other studies (Shalek *et al*, 2014; Xue *et al*, 2015; Muldoon *et al*, 2020).

## Discussion

Transcriptional bursting is an important process affecting many biological processes, but it has not been extensively studied for

endogenous NF-κB targets, including cytokines that are vital to the inflammatory response. Here, we explored changes in transcriptional bursting in response to the inflammatory cytokine TNF in T cells. We found that TNF can modulate either burst frequency or burst size depending on basal histone acetylation and regulation of RNAPII pausing. Using a small molecule inhibitor, we confirmed that altering basal histone acetylation before TNF stimulation modulated bursting behavior by reducing burst size and, in the case of *Tnf*, increasing burst frequency. Finally, we used mathematical modeling to show that TNF positive feedback can more efficiently amplify the skewed single-cell distribution that results from TNF-mediated increases in burst size as compared to a distribution of the same mean but different noise that results from a burst frequency increase. This suggests a possible biological consequence of TNF activating transcription via burst size that motivates further study.

We found that TNF primarily increased transcription by increasing burst size, which resulted in skewed, long-tailed mRNA distributions that are generally marked by large increases in Fano factor. In contrast, transcription factor-mediated increases in burst frequency result in less skewed distributions with lower cell-to-cell heterogeneity. Increases in burst frequency in response to transcription factor stimulation have been more commonly observed than increases in burst size (Li *et al*, 2018; Chen *et al*, 2019; Friedrich *et al*, 2019). Notably, most of these examples analyzed cellular processes for which it is important that most or all cells in a population respond to a stimulus with similar levels of gene expression such as the DNA damage response (Friedrich *et al*, 2019) or the circadian response to light (Li *et al*, 2018). In contrast, for processes where highly skewed single-cell responses might be beneficial, as could be the case for inflammatory signaling, stimulus-induced burst size increases may be more common. Long-tailed distributions with a few outliers far above the population mean have been shown to be important for regulating inflammatory signaling at the levels of single-cell transcription (Shalek *et al*, 2014) and cytokine secretion (Xue *et al*, 2015; Muldoon *et al*, 2020). Burst size regulation was also observed in response to Notch signaling, which is active in both embryonic development and maintenance of the germline stem cell niche (Falo-Sanjuan *et al*, 2019; Lee *et al*, 2019). Creating skewed distributions in transcription between cells that must follow different trajectories such as proliferation vs differentiation might help ensure that cells do not easily cross over to the other behavior. Burst size regulation of the growth factor *Ctgf* by multiple stimuli has been proposed to provide appropriate responses to different stimuli that require transient or sustained responses (Molina *et al*, 2013), which would support a role for burst size regulation in immune signaling.

We previously studied latent-but-inducible HIV LTR promoters integrated into Jurkat T cells (which are positively regulated by NF-

κB) and found that basal histone acetylation was associated with differences in TNF activation of transcriptional bursting (Wong *et al*, 2018). Using the small molecule histone deacetylase inhibitor TSA, we demonstrated that by increasing basal histone acetylation with TSA pretreatment, TNF activation of HIV changed from increasing burst frequency to increasing burst size. Here, we showed the opposite trend: By reducing basal histone acetylation with A-485 pretreatment, TNF activation of *Tnf* changed from increasing burst size to increasing burst frequency (Fig 5). Thus, the link between basal histone acetylation and TNF-stimulated transcriptional bursting appears to be causal for NF-κB target genes.

We also found that accumulation of promoter-proximal paused RNAPII was associated with TNF-mediated increases in burst size both at endogenous and HIV promoters (Wong *et al*, 2018). However, our studies with the small molecule JQ1, which inhibits pause release, were inconclusive. RNAPII promoter-proximal pausing occurs throughout the mammalian genome, especially at signal-responsive promoters (Adelman & Lis, 2012). Paused RNAPII primes a promoter to rapidly respond to an elongation signal, bypassing the need to recruit a new RNAPII subunit. Thus, the observation that accumulation of RNAPII is associated with burst size increases and more skewed transcript distributions warrants further study.

To model transcriptional bursting, we used the random telegraph model with one productive and one unproductive promoter state, and our results were largely consistent with our calculations of burst size and burst frequency based on the distribution moments. This simple promoter model sufficiently captured differences in transcriptional bursting "modes" following TNF treatment in Jurkat T cells that was a main focus of our study. However, two recent studies analyzing transcriptional bursting in response to stimulation of NF-κB by TNF or LPS reported somewhat different results (Bagnall *et al*, 2020; Zambrano *et al*, 2020). Bagnall *et al* studied activation of *Tnf* and *Il1b* following LPS stimulation in macrophages and found gene-specific mean-noise trends for *Tnf* vs *Il1b*; while a two-state promoter model was sufficient to reproduce *Tnf* distributions, a three-state model with an additional unproductive (or "refractory")

state was required to fit *Il1b* distributions. Zambrano *et al* similarly demonstrated that a promoter model with a third refractory state, combined with variability in upstream NF-κB signaling, was necessary to explain their observation of a subset of "first responder" cells that produced higher levels of *Nfkbia*, *Tnf*, and HIV in HeLa cells (Zambrano *et al*, 2020). We have previously demonstrated that variability in upstream NF-κB signaling is correlated with transcript levels in individual cells for the targets in our study even though they exhibit variations in noise (Wong *et al*, 2019), and thus, we do not think our results are inconsistent. We expect that most of the differences in our observations are due to the fact that mRNA levels of our targets are approximately an order of magnitude lower than in these other studies (mRNA ~ $10^1$ vs ~ $10^2$). We expect that complex model configurations will be necessary to reproduce cell-to-cell heterogeneity of endogenous NF-κB targets across cell types, stimulations, targets, and levels of expression.

HIV encodes its own positive feedback mediator, the protein Tat, which leads to amplification and viral activation in long-tailed distributions that stem from burst size increases but not from burst frequency increases (Wong *et al*, 2018). Because TNF also positively regulates its own expression via extracellular signaling, we used mathematical modeling to explore whether the shape of the single-cell *Tnf* mRNA distribution might be related to biological function similar to what we observed for HIV. As with HIV, we found that positive feedback more efficiently amplified transcript distributions resulting from burst size increases as compared to those resulting from burst frequency increases (Fig 6). However, our model has limitations. We phenomenologically modeled exogenous TNF stimulation with a time-dependent curve, rather than simulating NF-κB signaling, which would be affected by transcriptional noise from other targets such as the negative regulators *Nfkbia* and *Tnfaip3*. Moreover, we only accounted for autocrine signaling in our model and did not consider how the TNF produced by one cell might affect neighboring cells, which likely plays a major role in regulating immune signaling. Overall, our study motivates additional work to explore how transcriptional bursting in inflammatory gene expression functionally shapes the population immune response.

# Materials and Methods

**Reagents and Tools table**

| Reagent or resource | Reference or source | Identifier or catalog number |
|---|---|---|
| **Experimental model** | | |
| Jurkat T cells, clone E6-1 | ATCC | TIB-152 |
| **Antibodies** | | |
| Anti-histone H3 rabbit polyclonal | Abcam | ab1791 |
| Anti-acetyl-histone H3 rabbit polyclonal | Millipore | 06-599 |
| Anti-NF-κB p65 rabbit monoclonal | Cell Signaling Technology | 8242 |
| Anti-RNPII N-20 rabbit polyclonal | Santa Cruz Biotech | sc-899 |
| Anti-ser5-p RNPII rabbit polyclonal | Abcam | ab5131 |
| Anti-ser2-p RNPII rabbit polyclonal | Abcam | ab5095 |
| Anti-NELF-E H-140 rabbit polyclonal | Santa Cruz Biotech | sc-32912 |

**Reagents and Tools table**  (continued)

| Reagent or resource | Reference or source | Identifier or catalog number |
|---|---|---|
| Anti-acetyl-histone H3 rabbit monoclonal | Cell Signaling Technology | 8173 |
| Anti-RNPII NTD rabbit monoclonal | Cell Signaling Technology | 14958 |
| Anti-ser5-p RNPII rabbit monoclonal | Cell Signaling Technology | 13523 |
| Anti-TNF mouse monoclonal | eBioscience | 14-7348-81 |
| Anti-mouse IgG goat polyclonal, conjugated with Alexa Fluor 647 | Thermo Fisher Scientific | A-21235 |
| **Chemicals, peptides, and other reagents** | | |
| Roswell Park Memorial Institute 1640 medium (RPMI) | Thermo Fisher Scientific | 11875119 |
| Fetal bovine serum (FBS) | Atlanta Biologicals | S11150 |
| Penicillin-Streptomycin | Thermo Fisher Scientific | 15140122 |
| Human tumor necrosis factor-alpha (TNF) | PeproTech | 300-01A |
| A-485 | Structural Genomics Consortium | 6387 |
| JQ1 | Tocris | 4499 |
| Brefeldin A | BioLegend | 420601 |
| Sodium chloride (NaCl) | Thermo Fisher Scientific | 7647-14-5 |
| Tris-EDTA, pH 8.0 | Thermo Fisher Scientific | AM9858 |
| Cell-tak | Corning | 354240 |
| Lab-Tek #1.0 8-well chambered coverglass | Thermo Fisher Scientific | 155411 |
| µ-Slide 8-well glass bottom coverslip | Ibidi | 80827 |
| 70% Ethanol, RNase-Free | Thermo Fisher Scientific | 15420665 |
| 20× saline-sodium citrate (SSC) | Thermo Fisher Scientific | AM9763 |
| Formamide | Thermo Fisher Scientific | AM9342 |
| 50% Dextran Sulfate | Amresco | E516 |
| VectaShield | Vector Labs | H-1000 |
| 37% formaldehyde | Sigma-Aldrich | 818708 |
| 10× Phosphate Buffered Saline (PBS) | Thermo Fisher Scientific | AM9625 |
| Tween-20 | Thermo Fisher Scientific | 9005-64-5 |
| Bovine Serum Albumin (BSA) | Sigma | B4287 |
| Hoechst 33342 | Thermo Fisher Scientific | H21492 |
| cOmpleteTM protease inhibitor cocktail | Roche | 11836170001 |
| PureProteome protein G magnetic beads | Millipore | LSKMAGG10 |
| Salmon sperm DNA solution | Thermo Fisher Scientific | 15632011 |
| RNAse, DNAse-free | Roche | 11119915001 |
| Proteinase K | New England BioLabs | P8107S |
| EDTA | American Bio | AB00502-01000 |
| Lithium Chloride (LiCl) | Sigma-Aldrich | 7447-41-8 |
| IPEGAL CA630 | Sigma-Aldrich | I8896 |
| Glycine | American Bio | AB00730-01000 |
| Agarose | Lonza | 50074 |
| 20% SDS solution | American Bio | AB01922-00500 |
| Deoxycholic acid | Millipore | 302-95-4 |
| SYBR safe | Invitrogen | S33102 |
| Actinomycin-D | Millipore | 114666 |
| Sodium bicarbonate (NaHCO$_3$) | Alfa Aesar | 144-55-8 |
| Triton X-100 | American Bio | AB02025-00500 |
| SYBR Green supermix | Bio-Rad | 1725271 |

**Reagents and Tools table**   (continued)

| Reagent or resource | Reference or source | Identifier or catalog number |
|---|---|---|
| **Oligonucleotides** | | |
| Primers for RT–qPCR and ChIP-qPCR | Yale School of Medicine Keck Oligonucleotide Synthesis | Sequences available in Dataset EV2 |
| smFISH probes | Biosearch Technologies | Sequences available in Dataset EV2 |
| **Commercial Kits** | | |
| Upstate EZ-Magna ChIP | Millipore | 17-10086 |
| QIAQuick PCR purification kit | Qiagen | 28104 |
| RNeasy mini kit | Qiagen | 74104 |
| BD Cytofix/Cytoperm | BD Biosciences | 554714 |
| **Software and algorithms** | | |
| MATLAB 2016b, 2019b | MathWorks | |
| FISH-quant | Mueller *et al* (2013) and Tsarnov *et al* (2016) | |
| Mathematica 12 | Wolfram | |
| Prism 7 | GraphPad | |
| FlowJo | FlowJo, LLC | |
| NFsim | Sneddon *et al* (2011) | |

## Methods and Protocols

### Cell culture and pharmacological treatments

Jurkat T-cell clone E6-1 was obtained from ATCC. Jurkat cells were cultured in Roswell Park Memorial Institute 1640 (RPMI) medium (Thermo Fisher Scientific). All media was supplemented with 10% fetal bovine serum (Atlanta Biologicals), 100 U/ml penicillin, and 100 µg/ml streptomycin (Thermo Fisher Scientific). Cells were maintained in 5% $CO_2$ at 37°C and were never cultured beyond passage 20. Cells were grown to at least 500,000 cells/ml before treatment with 20 ng/ml recombinant human tumor necrosis factor α (TNF; Pepro-Tech), 300 nM A-485 (Structural Genomics Consortium), 62.5 nM JQ1 (Tocris), or Brefeldin A (diluted as directed; BioLegend).

### RT–qPCR

Total RNA was purified with the RNeasy Mini kit (Qiagen), including an on-column DNase treatment. cDNA was synthesized using Super-Script III reverse transcriptase (Thermo Fisher Scientific) and dT oligo primer. cDNA was diluted in nuclease-free water and quantified using SsoAdvanced Universal SYBR Green Supermix on a CFX Connect Real-Time System (Bio-Rad) with the following amplification scheme: 95°C denaturation for 90 s followed by 40 cycles of 95°C for 15 s, 60°C annealing for 10 s, and 72°C elongation for 45 s with a fluorescence read at the end of each elongation step. This was followed by a 60–90°C melt-curve analysis with 0.5°C increments to confirm product specificity. All samples were normalized to the house-keeping gene *Gapdh*. To calculate decay rates in Jurkat cells, we performed qRT–PCR after a 1-h TNF treatment followed by 10 µg/ml actinomycin-D treatment for varying times for *Nfkbia*, *Tnfaip3*, *Tnf*, and *Il8* (Appendix Fig S2A). All primer sequences are listed in Dataset EV2.

### smFISH probe design, hybridization, and imaging

The probe sets targeting *Nfkbia*, *Tnfaip3*, and *Il8* (Lee *et al*, 2014) and *Tnf* (Bushkin *et al*, 2015) were previously described. The probe sets targeting *Il6* and *Csf2* were designed using the Stellaris® RNA FISH Probe Designer (Biosearch Technologies, Inc., Petaluma, CA) available online (www.biosearchtech.com). All mRNAs were hybridized with Stellaris RNA FISH Probes labeled with Fluorescein (*Nfkbia*, *Il8*, and *Csf2*) or Quasar 670 (*Nfkbia*, *Tnfaip3*, *Tnf*, and *Il6*; Biosearch Technologies, Inc.) following the manufacturer's instructions. Briefly, Jurkat cells were treated under indicated conditions and then plated onto Cell-Tak (Corning) coated Lab-Tek #1.0 8-well chambered coverglass (Thermo Fisher Scientific) or µ-Slide 8-well glass-bottom coverslip (Ibidi). Cells were fixed in 3.7% formalde-hyde (Thermo Fisher Scientific) for 10 min and then permeabilized overnight in 70% ethanol (Fisher Scientific). Cells were hybridized for 12 h overnight with the following probe set specific conditions: 250 nM probe for *Tnf*/*Il6*/*Csf2* in 2× SSC (Thermo Fisher Scientific) with 10% formamide (Thermo Fisher Scientific) and 100 mg/ml dextran sulfate (Amresco) at 37°C, 50 nM probe for *Tnfaip3* in 2× SSC with 10% formamide and 80 mg/ml dextran sulfate at 37°C, 250 nM probe for *Nfkbia* in 2× SSC with 12% formamide and 100 mg/ml dextran sulfate at 37°C, and 250 nM probe for *Il8* in 2× SSC with 10% formamide and 100 mg/ml dextran sulfate at 25°C. For multiplex smFISH targeting *Nfkbia* and *Tnf*, probe concentrations were kept the same as for single gene smFISH and hybridization buffer for *Tnfaip3* was used. After hybridization, cells were washed twice with 2× SSC and 10% formamide, counterstained with 100 ng/ml Hoechst 33342 (Thermo Fisher Scientific) for 15 min, and immersed in VectaShield mounting media (Vector Labs). Cells hybridized with *Nfkbia*, *Tnfaip3*, *Tnf*, and *Il6* probes were imaged on an Axio Observer Zi inverted microscope (Zeiss) with an Orca Flash 4.0 V2 digital CMOS camera (Hamamatsu) and a 100× APO oil objective (NA 1.4, Zeiss). Cells hybridized with *Il8* and *Csf2* probes were imaged on a Nikon Eclipse Ti spinning disk confocal microscope (Yokogawa CSU-W1 spinning disk) with an Andor iXon Ultra888 EMCCD camera (Andor Technology) and a plan apochromatic 100× oil objective (NA 1.45, Nikon) after

identification of regions of interest within the field of view with high cell numbers. In all cases, Z-stacks of 30–80 images with 0.3 μm intervals were acquired. To avoid experimenter bias in selection of cells to image, only nuclear signal and not smFISH probe signal was used to select fields of view for imaging. All probe sequences are listed in Dataset EV2.

### smFISH image analysis

We quantified mRNAs in individual cells using FISH-Quant in MATLAB R2016B (Mathworks Inc.) (Mueller *et al*, 2013; Tsarnov *et al*, 2016). Cells were manually identified and outlined, with overlapping cells, cells partly in the field of view, and multinucleated cells excluded from analysis. Nuclei were initially outlined using FISH-Quant's "Detect nucleus" feature and then manually edited if necessary (in the cases of dim nuclei or nuclei that were very close, as these were challenging for the software to resolve in our images). Images of all genes were filtered using the Dual Gaussian filtering method in FISH-Quant with the default Kernel size settings: first, a large Gaussian Kernel (5 pixels) was used to blur the image for background subtraction, and then, a small Gaussian Kernel (0.5 pixels) was used to enhance small features in the background subtracted image (Appendix Fig S1). After image filtering, intensity thresholds to distinguish mRNA spots from background by identifying local maxima were determined by comparing the outputs of different thresholds to visually derived counts for both high and low expressing cells in addition to analysis of unstained control cells providing a minimum threshold. The remaining images were then processed in batch. Pre-detection intensity thresholds and detections settings varied with the type of microscope used for imaging, the fluorescent label on the probes, the specific probe set being used, and the experimental condition.

### Chromatin immunoprecipitation

Chromatin immunoprecipitation was performed using the Upstate EZ-Magna ChIP kit (Millipore). Briefly, 5 million cells per condition were fixed in 1% formaldehyde (Sigma) for 10 min, after which excess formaldehyde was quenched with 10× glycine at room temperature. Cells were washed three times with ice cold PBS and then lysed in 300 μl of 1% SDS lysis buffer with protease inhibitor cocktail (Roche). Lysates were sonicated with a Diagenode Bioruptor Plus with the following settings: 30 min of 30 s ON/30 s OFF at high power in a 4°C water bath. Sheared DNA was run on a 1% agarose gel (Lonza) to verify that sheared DNA was between 100 and 1,000 bp. Samples were pre-cleared with PureProteome Protein G magnetic beads (Millipore) at 4°C and 5% of each sample was aliquoted as a percent input control. Samples were incubated with antibody at manufacturers' recommended concentrations overnight at 4°C. PureProteome beads were added and incubated for 1 h at 4°C. Beads were washed once each with low salt, high salt, and LiCl immune complex wash buffers, then washed twice with TE buffer, and then eluted with elution buffer at room temperature. Crosslinks were reverse by incubating samples with NaCl overnight at 65°C. DNA was purified using the QIAQuick PCR Cleanup kit (Qiagen). DNA was quantified using quantitative PCR using SsoAdvanced Universal SYBR Green Supermix on a CFX Connect Real-Time System (Bio-Rad). qPCR was run in triplicate, and melt curves were run to confirm product specificity. All primer sequences are listed in Dataset EV2.

### Flow cytometry

Cells were prepared for intracellular cytokine staining to detect TNF production using the BD Cytofix/Cytoperm kit (BD Biosciences). Briefly, 100,000 cells per condition were treated with TNF and Brefeldin A (BioLegend), then washed with PBS, and fixed with Fix/Perm for 20 min at 4°C. Fixed cells were washed twice with 1× Perm/Wash. Cells were stained with 4 μg/ml (1:125 dilution) anti-TNF (eBioscience # 14-7348-81) for 1 h at 4°C, washed twice with 1× Perm/Wash, and then stained with 10 μg/ml (1:200 dilution) anti-mouse-AlexaFluor647 (Thermo Fisher Scientific A-21235) for 1 h at 4°C. All data were acquired on an Attune NxT Flow Cytometer (Thermo Fish Scientific) analyzed with FlowJo (FlowJo, LLC).

### Fitting the two-state model

Maximum-likelihood estimation (MLE) was used to select burst frequency ($k_a$) and burst size ($b = k_t/k_i$) parameters that best fit the measured mRNA distributions to the full analytical solution to the two-state stochastic gene expression model (Peccoud & Ycart, 1995). Although this is a steady-state solution, we use it here to approximate how TNF affects transcriptional bursting (Wong *et al*, 2018). We assumed that the two alleles for each gene were independent and that bursting was sufficiently infrequent such that bursting events were unlikely to overlap, allowing a reasonable estimate of burst size and an upper bound on the estimate of burst frequency by modeling transcription from a single allele. MLE was performed as numerical minimization over the negative log-likelihood function defined over the probability density function (pdf) given the observed experimentally determined RNA distributions for each condition using the method of moments. As previously reported, mRNA distributions are not sufficient to independently determine the promoter inactivation rate $k_i$ and the transcription rate $k_t$. Using a previously described method (Raj *et al*, 2006; Dey *et al*, 2015), we held the transcription rate $k_t$ constant across all conditions and reported $b$. Sensitivity analysis of the $k_t$ value for each gene suggested that our results are largely independent of the $k_t$ value chosen for each gene (Appendix Fig S2B). MLE was implemented using custom code in Mathematica 8 (Wolfram Inc.) as previously described (Dey *et al*, 2015). The model was fit to smFISH distributions from combined replicates except for the *Nfkbia* TNF 1-h time point. The model was unable to produce a fit for the combined dataset and thus replicates were fit individually. An example fit is included in Fig EV3, but the burst size and burst frequency were not reported due to this discrepancy.

### Statistical analysis

To compare conditions for cell-population measurements, the *f*-test was first applied to determine whether datasets were heteroscedastic, and then the Student's or Welch's *t*-test was applied as appropriate. A Dunnett's *t*-test was used for multiple comparisons. Regression and correlation analyses was performed in Prism (GraphPad). All tests were performed with an alpha value of 0.05. All smFISH experiments included a sufficient number of cells to characterize the transcript distributions ($n > 100$ cells). A summary of all experimental conditions, biological replicates, and total cell numbers collected by smFISH is included in Table 1. The 95% confidence intervals (CIs) on all descriptive statistics of RNA distributions were estimated from the 2.5% and 97.5% quantiles of bootstrapped copy number counts per cell as previously described

**Table 1. Summary of smFISH experiments.**

| Target | Basal | | 1 h TNF | | 2 h TNF | | 4 h TNF | |
|---|---|---|---|---|---|---|---|---|
| | Reps | Total cells | Reps | Total cells | Reps | Total cells | Reps | Total cells |
| *Nfkbia* | 3 | 611 | 3 | 628 | 2 | 356 | NA | NA |
| *Tnfaip3* | 3 | 529 | 3 | 352 | 1 | 542 | NA | NA |
| *Tnf* | 3 | 839 | 3 | 858 | 3 | 616 | 1 | 219 |
| *Il8* | 1 | 248 | 1 | 477 | 1 | 426 | NA | NA |
| *Il6* | 3 | 632 | NA | NA | 3 | 599 | 3 | 661 |
| *Csf2* | 1 | 167 | NA | NA | 1 | 232 | 1 | 269 |

| Target | A-485 4 h →TNF 1 h | | TNF + JQ1 1 h | | TNF + BFA 2 h | | | |
|---|---|---|---|---|---|---|---|---|
| | Reps | Total cells | Reps | Total cells | Reps | Total cells | | |
| *Tnfaip3* | 2 | 373 | 2 | 348 | NA | NA | | |
| *Tnf* | 2 | 401 | 1 | 216 | 1 | 148 | | |

(Dey *et al*, 2015). For those samples for which we had sufficient replicates, we confirmed that the 95% CIs matched the error estimated by calculating the SD from three biological replicates (Appendix Fig S3). 95% confidence intervals on fit burst frequency and size parameters were estimated from the log-likelihood function assuming asymptotic normality of the estimates and using 1.92 log-likelihood ratio units as previously described (Dey *et al*, 2015). The difference between two quantities was inferred to be significant ($P < 0.05$) if the 95% CI's were not overlapping (Schenker & Gentleman, 2001).

### Transcription model development

We modified an existing two-state bursting parameter model (Wong *et al*, 2018). We modeled transcription as a promoter that transitions from an "OFF" state to an "ON" state, and vice versa, with rate constants, $k_a$ and $k_i$, respectively. In the "ON" state, mRNA is produced at the rate $k_m$ and degraded at a rate of $g_m$. This rate was modulated by time via a fitted burst size curve (see below). The mRNA produces TNF protein at the rate $k_p$, is exported out of the cell at a rate $k_{ex}$, and degraded at a rate of $g_p$. TNF is known to positively feedback onto its own production, and so a feedback loop was introduced into the model to increase the rate of mRNA production, $k_m$, as TNF increased. The reactions governing this model, along with accompany rate constants, are described in Table 2. We modeled only one promoter activation event (i.e., a single allele),

consistent with our fitted estimates of burst frequency, and after confirming via simulations that bursts from two alleles rarely overlap (Appendix Fig S4).

This model is represented by the following system of ordinary differential equations:

$$\frac{d[PromotorOff]}{dt} = -ka*[PromotorOff] + ki*[PromotorOn] \quad (1)$$

$$\frac{d[PromotorOn]}{dt} = ka*[PromotorOff] - ki*[PromotorOn] \quad (2)$$

$$\frac{d[mRNA]}{dt} = \left\{ 1 + \frac{A*[TNF(outside)]}{K + [TNF(outside)]} \right\} * k_t * PromotorOn - gm*mRNA \quad (3)$$

$$\frac{d[TNF(inside)]}{dt} = ap*[mRNA] - (gp + k_{ex})*[TNF(inside)] \quad (4)$$

$$\frac{d[TNF(outside)]}{dt} = k_{ex}*[TNF(inside)] - gp*[TNF(outside)]. \quad (5)$$

To simulate treatment with exogenous TNF, we altered our burst size dynamically to reflect that TNF treatment increases experimental burst size in a time-dependent curve. Our BFA experiment showed that *Tnf* mRNA is lower at 2 h when extracellular feedback

**Table 2. Model parameters**

| Reaction | Parameter (units) | Values | Source |
|---|---|---|---|
| *Promotor On →Promotor Off* | $k_i$ (h$^{-1}$) | 15 | Experimental derivation from smFISH data |
| *Promotor Off→Promotor On* | $k_a$ (h$^{-1}$) | 1.3 | Experimental derivation from smFISH data |
| *Promotor On→mRNA* | $f_m*k_t$ (h$^{-1}$) | $k_i * b$ | Calculated from fitted burst size equation ($b = k_m/k_i$) |
| $fm = 1 + \frac{A*[TNF(outside)]}{K + [TNF(outside)]}$ | *K, A* | 500, 25 | Parameter scan (Fig EV5D) |
| *mRNA→TNF (inside)* | $k_p$ (h$^{-1}$) | 0.75 | Estimated from (Caldwell *et al*, 2014) |
| *TNF (inside)→TNF(outside)* | $k_{ex}$ (h$^{-1}$) | 18 | (Paszek, *et al*, 2010) with assumptions from (Lee, *et al*, 2014) |
| *TNF →∅* | $g_p$ (h$^{-1}$) | 0.36 | Estimated from mRNA degradation (1/3 of $g_m$) |
| *mRNA→∅* | $g_m$ (h$^{-1}$) | 1.09 | Experimental derivation |

was blocked (Fig EV5A). We performed a weighted Gaussian curve to the burst sizes inferred from our experimental smFISH distributions for basal, TNF alone (1, 4 h), and TNF + BFA (2 h) in MATLAB (Fig EV5C). This phenomenological equation reflects TNF's mechanistic activation of NF-κB in the absence of positive feedback, and further promotion of transcription. This equation allows burst size to change dynamically with time, and alter $k_t$, while burst frequency is assumed to remain constant. We note that this is an approximation that is based on experimental observations that TNF modulates burst size more than burst frequency (Fig. 3C); however, it does not fully reflect the data in the presence of BFA (Fig EV5A). The overall rate influencing transcription can also be altered by positive feedback ($f_m$). To replicate scenarios without positive feedback, amplification A was set to 0.

To stochastically simulate TNF protein and mRNA transcript production over time, we used network-free stochastic simulator (NFSIM) (Sneddon *et al*, 2011). All analysis and plots were done in MATLAB R2019B (MathWorks, Inc.).

### Steady-state analysis

To understand how the system behaves under basal conditions, we assumed equilibrium for the above equations. First, we examined promotor dynamics, and solved for steady-state. By solving equation 1 and 2 at steady-state, and setting

$$[PromotorOff] = 1 - [PromotorOn] \qquad (6)$$

we derive the following:

$$[PromotorOn] = \frac{k_a}{k_i} * [PromotorOff]$$

$$[PromotorOn] = \frac{k_a}{k_i + k_a} = B \qquad (7)$$

We can then examine TNF concentrations inside and outside the cell (equations 4 and 5), by deriving the following:

$$k_{ex} * [Protein(inside)] = gp * [Protein(outside)]$$

$$ap * [mRNA] = (gp + k_{ex}) * [Protein(inside)]$$

$$[Protein(outside)] = \left(\frac{ap}{gp + k_{ex}}\right) * \left(\frac{k_{ex}}{gp}\right) * [mRNA] = C * [mRNA].$$

$$(8)$$

Finally, we use EQ3, EQ7, and EQ8 to solve for mRNA values under steady-state conditions, following:

$$gm * [mRNA] = [PromotorOn] * k_t * \left\{ 1 + \frac{A * [Protein(outside)]}{K + [Protein(outside)]} \right\}$$

$$\frac{gm * C}{B * k_t} * [mRNA]^2 + \left\{ \frac{gm * K}{B * k_t} - C - A * C \right\} * [mRNA] - K = 0. \qquad (9)$$

Using this equation, we explored how the parameter space affects mRNA concentration before TNF treatment. By varying feedback parameters—the amplification rate, and the K half-max—we recreated regions that matched basal *Tnf* mRNA conditions. To

understand how the unbounded feedback parameters influenced the model upon TNF treatment, we ran 2D parameter scans altering feedback parameters. Two parameters were chosen that qualitatively reproduced the time course of TNF-activation experiments under deterministic simulation of the model (Fig EV5D) and replicated steady-state values of basal mRNA.

To explore how changes in burst size and burst frequency influenced phenotypic outcomes, we stochastically simulated 1,000 cells using NFSIM, altering the parameters $k_i$ and $k_a$. Four representative parameter combinations were chosen, each with an average of 10 mRNA transcripts per cell at 1 h (Fig 6F and G).

## Data availability

Quantitative smFISH measurements presented in the main figures are provided as figure source data, labeled "Dataset EV1". All PCR primer and smFISH probe sequences used for this work are provided in "Dataset EV2". Code to reproduce the mathematical model is available at https://github.com/elisebullock/tnftwostate.

**Expanded View** for this article is available online.

### Acknowledgements

We thank the reviewers for a careful and constructive reading of our manuscript. We thank Dr. Nadya Dimitrova for advice on chromatin immunoprecipitation, and Dr. Valerie Horsley and Dr. Yannick Jacob for use of laboratory equipment. We thank Dr. Sanjay Tyagi for sharing TNF smFISH probe sequences. We thank the Structural Genomics Consortium for providing the small molecule A-485. This work was funded by the National Science Foundation (CBET-1454301 to K.M.-J.), and the NIH (1R01-GM123011 to K.M.-J.). V.L.B. was supported by NIH predoctoral training grants in virology (5T32AI055403-12 and 5T32AI055403-13). V.C.W. was supported by NIH predoctoral training grants in genetics (2T32GM007499-36, 5T32GM007499-34, and 5T32GM007499-35). M.E.B. was supported by NIH grant 1T32EB019941.

### Author contributions

Study conception and experiment design: VLB, VCW, SG, and KM-J. Experiments: VLB and VCW. Mathematical and stochastic modeling: MEB Data analysis: VLB, VCW, MEB, and KM-J. Figure preparation and manuscript writing: VLB, MEB, and KM-J. Manuscript editing: All authors. Funding and research supervision: KM-J.

### Conflict of interest

The authors declare that they have no conflict of interest.

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
