## [Review Process File · Molecular Systems Biology]

TNF stimulation primarily modulates transcriptional burst size of NF- κ B-regulated genes

Victor Bass, Victor Wong, M. Bullock, Suzanne Gaudet, and Kathryn Miller-Jensen
DOI: [10.15252/msb.202010127](https://doi.org/10.15252/msb.202010127)

Corresponding author(s): Kathryn Miller-Jensen (kathryn.miller-jensen@yale.edu)

Review Timeline:

Submission Date:	15th Nov 20
Editorial Decision:	21st Dec 20
Revision Received:	29th Mar 21
Editorial Decision:	10th May 21
Revision Received:	3rd Jun 21
Accepted:	7th Jun 21

Editor: Maria Polychronidou

Transaction Report:

Thank you again for submitting your work to Molecular Systems Biology. We have now heard back from the four referees who agreed to evaluate your study. Overall, the reviewers acknowledge that the study seems potentially interesting. However, they raise a series of concerns, which we would ask you to address in a major revision.

I think that the recommendations of the referees are rather clear and there is therefore no need to repeat the comments listed below. A particularly fundamental issue, raised by multiple reviewers, is the need to include further experimental validations of the proposed TNF-induced positive feedback loop. Moreover, as the reviewers point out, several of the presented conclusions need to be better supported by adequate number of replicates, quantifications, statistical support and a clearer description of the methodological details.

Please let me know in case you would like to discuss in further detail any of the issues raised. All issues raised by the referees would need to be satisfactorily addressed. As you may already know, our editorial policy allows in principle a single round of major revision and it is therefore essential to provide responses to the reviewers' comments that are as complete as possible.

On a more editorial level, we would ask you to address the following.

Reviewer #1:

This is an interesting manuscript that studies NFkB target gene expression and its relation to underlying noise in T cells (Jurkat cell line). The authors chose an important pathway and cell type, relevant to medical problems and immunology, and studied single cell and population based gene expression variability and supplemented their experiments with modeling. The main conclusion of the paper is that TNF stimulation induces NFkB gene expression by increasing promoter burst size and not burst frequency. This is an interesting finding and is well supported by the analyses in the paper. Their second surprising conclusion is that TNF induced positive feedback can result in increasing cell to cell variability in gene expression. While this is suggested by their theoretical analyses, it is not supported by experiments. The TNF transcripts produced by the cells are very low (10-15 per cell) and it is not clear if this can lead to sufficient TNF production-release-and feedback by restimulating the cells. This should be explored experimentally before accepting this paper for publication. They can measure TNF production by the same cells upon TNF stimulation, for example by single cell ELISA (ELISPOT). Other main conclusion of the paper is that gene expression noise (variability) is inversely proportional to mean copy number of the transcripts expressed by the gene (first two figures), but this is not surprising. This is a simple consequence of poisson-like distributions, and was observed before. In conclusion, this paper would be a good candidate for publication in MSB if the authors can test that their positive feedback idea is supported by experiments (or not). If the experiments fail to show such strong TNF production, they can either caveat or remove the last portion of the findings which seem to be additional to the main finding anyway.

Reviewer #2:

This manuscript by Bass and colleagues describes the transcriptional bursting response of NF-kB target genes by using RNA FISH after TNFalpha stimulation. They conclude that immediate early response genes modulate their burst sizes in response to stimulation, in contrast to later responding genes that modulate their burst frequency. The authors link the increased burst sizes to Pol II pausing. The findings are interesting and novel, and the data is generally of good quality. However, data presentation and discussion are often confusing (see comments below), and the changes in pol II pausing as main cause of changes in burst size is not entirely convincing. The authors should address the points below to make their manuscript suitable for publication.

Major comments:

1. The authors mention that cell size does not contribute to the variability in mRNA expression levels. They then conclude that "shared sources of cellular variation are less important than gene-specific noise sources". This is an overstatement, since there are many other sources of cellular variation. For example, they ignore cell cycle progression, which is typically a very strong contributor to variability in gene expression. This should be addressed in their manuscript.
2. The claims around the effects of A485 and JQ1 are confusing.
 - "These changes were more pronounced for Tnf, for which A-485 pretreatment completely eliminated the long-tailed distribution of cells expressing high numbers of mRNA, consistent with its impact on Tnf Fano factor". I don't understand how the authors come to this conclusion when looking at Fig.5E and 5F. To me there is no visible difference between A485 and JQ1 on the figure.

- Fig.5C: changes in RNAPII Ser5P are rather small and do not scale with changes in burst sizes for the two genes. The authors should explain what they expect when they use JQ1 in terms of amount of paused PolII and how this relates to burst sizes
- Fig.5G:
 - a) "For Tnf, A485 pretreatment prior to TNF stimulation caused an increase in burst frequency without any change in burst size compared to the basal state (Fig. 5G, H)". I don't see the data of pretreatment prior to TNF stimulation anywhere in the figure, so I am confused.
 - b) Which panel corresponds to which gene ? It seems that A485 decreases burst sizes even more than JQ1, but the authors focus their conclusions on pausing in relationship with burst sizes.
- 3. The last part on the positive feedback loop is difficult to follow. The authors should improve the way they explain how their simulations in Fig.6B-E fit with the real data

Other comments:

- The authors mention chromatin accessibility while looking at histone acetylation. While this correlation might generally hold true, they should refrain from mentioning something they do not measure directly (chromatin accessibility), or even better, perform ATAC-seq to justify their claims.
- Fig.1J and 2D: The data presentation should be changed, it is very difficult to understand which datapoint corresponds to what.
- The author state at the bottom of page 5: "Average mRNA levels were inversely correlated with Ach3:H3 ratio.". This is confusing, I don't see this inverse correlation anywhere, and it seems to me that it should be the opposite.
- Fig.2A: The duration of the TNF treatment is not specified
- Page 9, top: they cite Fig.EV4 for distribution plots - do they mean Fig.EV3 ? Also they should show TNFa stimulated distributions for Nfkb1a mRNAs.
- Fig.4: A: which are the dashed ones ? B: what does the color code mean ?
- Page 11: "a large reduction in the number of cells expressing much higher than the mean for both genes (Fig. 5D)". Do they mean Fig.5E ?

Reviewer #3:

The paper by Bass et al analyses the effect of TNF stimulation on the variability of the NF- κ B-target genes. The authors demonstrate that TNF affects the chromatin and polymerase interactions at the specific gene promoter, which ultimately leads to changes in the respective mRNA distributions (and variability). Authors then use telegraph model to understand the modulation of transcriptional bursting in terms of burst size and frequency (in particular in resting cells vs cells stimulated with TNF). They demonstrate that TNF stimulation affects burst size as well as burst frequency in different gene subsets. Finally, they use a model of positive TNF feedback to evaluate the role of transcriptional bursting in the generation of cellular heterogeneity.

Overall, this is a timely subject and elements of the manuscript are performed to a high standard. I do appreciate the relatively large ChIP dataset and use of inhibitors. However, other elements in my opinion require more work. I have questions regarding the justification and application of the telegraph model and fitting protocols. I also believe that the paracrine feedback analysis is potentially interesting but should be validated by additional experimentation. Moreover, I do struggle to assess how robust are specific mRNA datasets, from reading the manuscript it is unclear whether data is replicated, how many cells are available, and how consistent are replicates (in terms of mean, CV, Fano factor and other estimates).

Specific comments:

Fig. 1.

Not clear from the text if the smFISH data is replicated, CIs are produced by bootstrapping. In G some of the CIs are missing. Are statistics calculated based on individual replicates show similar differences as those obtained from bootstrapping.

"The slope of this conserved mean-versus-noise trendline suggests non-Poissonian stochastic transcription rather than continuous transcription in inducible NF- κ B targets (Dar et al, 2016; Singh et al, 2010)." For clarity, can the Poisson model be plotted (i.e. $CV^2 \sim 1/\mu$) on the graph. This will highlight how much more noise there is in the data.

Fig 2.

"We found that for the NF- κ B targets that increased mean without a significant reduction in noise, we observed that in some cases they moved outside the basal trendline, especially at 2 hours (Fig. 2D, left). In contrast, Il6 and Csf2 remained within the trendline of the basal measurements upon TNF treatment for 2 and 4 hours as noise decreased with an increased mean (Fig. 2D, right). Overall, this suggests that NF- κ B differentially regulates transcriptional noise at different target genes following TNF stimulation."

These panels are difficult to understand; can the authors highlight different genes, what are the specific conditions that do not follow the trend. The suggestion that heterogeneity may be altered somehow is rather vague. Recently, (Bagnall et al., 2020), demonstrated that NF-kappaB dependent genes in response to stimulation exhibit conserved linear mean-variance relationships, which describe how noise changes in response to stimulation/perturbation. The genes considered here should follow similar constraints (given linear mean-variance, there should be a nonlinear CV^2 -mean relationship for each gene), which can aid with interpretation of the data.

"We found no significant reductions in noise after normalizing for cell area (Fig. EV1C), in contrast to other targets for which single-cell mRNA expression has been shown to correlate with cell size (Bagnall et al, 2018; Padovan-Merhar et al, 2015). This lack of correlation with cell size suggests that, for these inflammatory gene targets, shared sources of cellular variation are less important than gene-specific noise sources." The statement somehow lacks precision, the cited paper looked at TNF and NFKBIA, which are the same genes.

Fig. 3:

There is no justification for using an intrinsic noise model here. Authors should be able to look at their smFISH data, quantify the level of intrinsic vs extrinsic noise, and determine whether the telegraph model is appropriate in this case. This is important because recent evidence suggests a substantial contribution of the extrinsic noise in the NF- κ B-dependent transcription (Zambrano et al, 2020), i.e. transcription kinetics appears to be pre-determined in a subset of cells.

"Using a previously described method (Rat et al., 2006. Skupskey et al., 2010), we held the transcription rate k_t constant across all conditions and reported b ."

There are more accurate methods that allow fitting all the parameters in question, especially those taking into the account temporal regulation and measured basal distributions- see for example (Gomez-Schiavon et al., 2017). The assumption that the k_t does not change, in particular between basal and induced conditions seems very strong. Authors imply that a basal state works in a different mode than the inducible state, therefore why k_t should be the same. For example, the TNF

mRNA fitting by (Bagnall et al., 2020) show in fact large changes in the k_t value between basal and stimulation conditions.

It is not clear is whether the model assumed two or one alleles, which will affect fitted parameter values (also model with 2 alleles might allow fitting *nfkbia* count data). Also, the authors use " k_a " as the estimator of the bursting frequency. This is only an approximation and assumes that the gene is in the 'bursty' regime, roughly $k_{off} \gg k_{on}$. Therefore, the authors should check if this assumption is actually satisfied with their data. Otherwise, a more generic estimator for frequency should be used $f = k_a * k_i / (k_a + k_i)$ (Nicolas et al., 2018).

Fig. 4

Are the changes (basal vs. stimulated) statistically significant?

Panel B needs an indication of the scale (i.e., blue-red levels). Relationship between CV/Fano factor and burst size/frequency highlighted in the legend should be explained explicitly

Fig. 5

"To perturb histone acetylation, we pre-treated Jurkat cells with the histone acetyltransferase (HAT) inhibitor A-485, a specific inhibitor of the HATs p300/CBP that are recruited by NF- κ B (Fig. 5A) (Gerritsen et al, 1997; Lasko et al, 2017). We found that treatment with A-485 for 4 hours decreased Ach3 levels at the *Tnfaip3* and *Tnf* promoters but did not significantly affect total H3 levels as measured by ChIP-qPCR (Fig. 5B)." Statistical analysis is performed (also in C), a pairwise t-test based on 2 replicates (as legend says). I would not deem this as an appropriate use of statistics.

Panel D- no indication whether the data was replicated, how many cells are available in this dataset, are means of replicates per condition consistent?

Authors state: "Overall, these inhibitors decrease TNF-stimulated transcription while differentially affecting mRNA distributions. " In fact, the distributions in E corresponding to both inhibitors appear to be quite similar to each other, as indicated by fitted bursting characteristics in F and EV4.

Panel G- labelling is missing, not sure were is TNF vs TNFAIP3. Authors state "Fitting mRNA distributions for TNF treatment following pre-treatment with A-485 to the theoretical pdf of the two-state model, we found that, in response to TNF treatment, A-485 decreased burst size for *Tnfaip3* while not affecting burst frequency. For *Tnf*, A485 pre-treatment prior to TNF stimulation caused an increase in burst frequency without any change in burst size compared to the basal state (Fig. 5G, H)." In fact panel B suggests that the burst size is decreased.

The authors try to draw generic conclusions based on 2 inhibitors and behaviour of two genes. I find it quite difficult to follow and accept, especially not knowing how robust this dataset is. A number of points is often made based on a correlation between a panel of genes and CHIP data.

Fig. 6.

As TNF can stimulate its own transcription, we included positive feedback from the newly produced TNF on the cell that produced it (simulating autocrine signaling) to explore its effects on cell-to-cell heterogeneity (Fig. 6A). We modelled the addition of exogenous TNF as a time-dependent change

in kt , the mRNA production rate, in order to match the change in burst size inferred from our smFISH distributions. "

Here the authors provide a theoretical study of the potential effect of the positive TNF feedback on the transcriptional bursting. I would expect that positive-feedback should modulate responses by contributing to the ongoing NF- κ B response/variability, which then modulates transcriptional bursting by affecting k_{on} , k_{off} , and kt rates (directly through the probability of NF- κ B binding and dissociation or indirectly by further changing chromatic state, as authors suggest). Therefore, the explicit assumption about the effect of feedback on the kt rate seems very strong. Especially that in Fig. 3 authors assume that this rate does not change between basal vs TNF stimulated conditions.

More importantly, the validation of these simulations is important. Does Jurkats produce a meaningful amount of TNF (in relation to the 20ng/ml stimulation dose) in the time scale of the experiment? Is there a difference in the mRNA distribution when the feedback (TNF or generic) is blocked experimentally, does it induce predicted changes to transcriptional bursting? I feel that without this validation the modeling is very speculative. Ultimately the modeling demonstrates that if we change burst size or frequency, we do affect the heterogeneity. But this is a property of the telegraph model. (Bagnall et al., 2020) demonstrates that the modulation to burst size and frequency follows relationships can be predicted by the gene-specific linear relationships. This might help to interpret the data presented in this manuscript.

Bagnall, J., Rowe, W., Alachkar, N., Roberts, J., England, H., Clark, C., Platt, M., Jackson, D.A., Muldoon, M., and Paszek, P. (2020). Gene-Specific Linear Trends Constrain Transcriptional Variability of the Toll-like Receptor Signaling. *Cell Syst* 11, 300-314 e308.

Gomez-Schiavon, M., Chen, L.F., West, A.E., and Buchler, N.E. (2017). BayFish: Bayesian inference of transcription dynamics from population snapshots of single-molecule RNA FISH in single cells. *Genome Biol* 18, 164.

Nicolas, D., Zoller, B., Suter, D.M., and Naef, F. (2018). Modulation of transcriptional burst frequency by histone acetylation. *Proc Natl Acad Sci U S A* 115, 7153-7158.

Reviewer #4:

Report for "TNF stimulation primarily modulates transcriptional burst size of NF- κ B-regulated genes"

The authors perform an analysis of the NF κ B transcriptional response following TNF stimulation in Jurkat cells. NF κ B is obviously an important transcription factor mediating inflammatory and stress responses. While the transcriptional response parameters and bursting properties of the HIV long terminal repeat (LTR) promoter have been fairly extensively studied, less is known about the responses of endogenous NF- κ B targets. It is well known that the analysis of transcriptional responses in mammalian cells at steady-state boils down to the description of the burst frequency and size, which can be done using live reporters (for example MS2) but also with smFISH even though the latter is less direct. Here the authors study transcriptional bursting at six endogenous NF- κ B target promoters before and after TNF stimulation, including both primary inflammatory genes (direct targets) as well as two secondary genes. Upon TNF stimulation increases, mean transcription of the direct targets increased while noise (CV) did not significantly increase, which is equivalent to the statement that stimulation primarily increases burst size while maintaining burst frequency. It is then argued that the increase in transcriptional burst size is linked to TNF-dependent regulation of RNAPII pausing. While the notion that activation of certain signaling

pathways transiently increases burst sizes has been described before, the specific results on NFkB transcriptional response are interesting and timely. There are methodological and presentation issues that need to be addressed, which will make the manuscript significantly stronger.

*** Major ***

1) The smFISH images and associated quantifications in 1F,E and 2A-C are quite atypical and seem of heterogeneous quality. In Figure 1 it appears that the FISH signal is essentially in the nucleus and it is unclear what a single mRNA spot looks like. Why is the background signal in the FISH channel so high, particularly in the nucleus? Is there a potential issue with the hybridization conditions or probe specificity, or is this reflecting the image filtering?

Concerning the quantifications, comparing Nfkb1a I18 and Csf2, it is not clear why these genes differ quite strongly in the number of counts per cell. In general the correspondence between the images and quantifications is difficult to grasp.

Similarly the comparison between Figure 1E and 2A is confusing. In 2A (Tnfrsf25), mRNA dots are clearly visible while this was not the case in 1E. I18, which clearly has the strongest signal in 2A, is reported with only max 10 mRNA counts.

Thus, the authors need to significantly improve their presentation and potentially analysis of the smFISH data. In particular more justification and supplementary material is needed, showing more and larger raw (unfiltered) images, and show how the quantification and counting of the mRNA in each cell was done. It is not enough to simply refer to the FISH-quant software.

2) Besides a general statement in the Methods (see next paragraph) it was hard to find the information on how the authors handled replicates of smFISH experiments for each Figure. There's a short statement in the Figure EV1 caption, but it should be included systematically. It should also be stated clearly what variability the bootstrap confidence intervals in 1G-I, 2B 5G-H are capturing. It seems that the data are mostly analyzed in a pooled manner, except in Figure EV1C. In the methods it is stated that "All smFISH experiments included a sufficient number of cells to characterize the transcript distributions ($n > 100$ cells) and results were confirmed with independent biological replicates." but it was difficult to understand what is meant by 'confirmed' for each experiment shown.

3) Panel J: "The slope of this conserved mean-versus-noise trendline suggests non-Poissonian stochastic transcription rather than continuous transcription". The interpretation of the CV - mean relationship could be strengthened. The fact that the relation is linear in log-log (or power law in the original variables) is interesting, but it's really the values of the slope and intercept that matter. Poisson noise corresponds to a slope of -1 and zero intercept. Therefore this results need to be interpreted with respect to the slope and intercept of the fitted line. For example, for the telegraph model the CV^2 equals $(b+1)/m$, where m is the mean and b the burst size. This is partially discussed in the section on burst sizes but could be started here.

4) There is also a statement on Figure 2D in the section on burst size "Interpreted in this way, the mean-versus-noise plots suggest TNF differentially regulates burst frequency and burst size across different NF- κ B target genes (Fig. 2D) (Sanchez and Golding, 2013)." This is interesting but is hard to understand without further explanation.

5) When introducing the random telegraph model as a way to analyze the noise, the authors should clearly state that this does not take into account extrinsic noise or other form of

heterogeneity. This is a very strong assumption and that should be justified (cell cycle states, microenvironment, etc). Departure of the data from this assumption can easily lead to wrong conclusions after fitting. How do the authors justify this approximation, in particular since it appears that there is heterogeneity in the proportion of responders? In principle, more complex models would be needed to take into account such heterogeneity.

6) Figure 4. The interpretation of an increased level of paused PolII relies on subtle differences between s5p and s2p that may not be large enough to make a strong point. It also depends on the location of the PCR primers, which is not discussed in the main text. It would be useful to measure RNPII in the gene body to confirm that the level of transcription is indeed low in the paused state.

7) A recent work by Zambrano et al (iScience 2020, PMID: 33083759) is briefly cited but it would be interesting to more extensively discuss it since this study also performed time courses of endogenous genes by smRNA FISH, as well as live analysis with an MS2 reporter, though in a different cell line. Also, the finding here are reminiscent of other systems where the activation of signaling transiently increases burst sizes, which should be mentioned (Molina et al. PMID: 24297917).

8) A general puzzle is that Figure 1 clearly show that the transcriptional action of the direct targets takes place at very short times (probably <1 hour if one were to look at nascent transcription). Yet, most experiments aimed at elucidating the mechanisms are done at 2 and 4 hours (Figure 5), which thus seems too late.

*** Minor ***

"Average mRNA levels were inversely correlated with Ach3:H3 ratio." Isn't Figure 1B showing the opposite? This was confusing as Ach3 is typically a mark that correlates positively with activity.

In Figure 2A and 2D, it is not specified what the time point is.

Figure 2D. It is not mentioned what the blue and orange dots represent, what are the criteria for coloring dots in blue or orange? Since 6 genes are measured, it was unclear why there are so many dots. The "caption and description in the text needs to be improved.

"The one exception was for Nfkbia, in which the theoretical pdf from the two-state model could not be accurately fit to the TNF-stimulated distributions. This is consistent with the high transcriptional rate induced by TNF that might be outside the limits of the two-state bursting model (Wilson et al, 2017)." This is unclear, there is not restriction on the transcription rate in the two-state model. A potential caveat of the fits is that the theoretical distributions are appropriate in steady-state (or quasi steady-state) situations, which is not the case here as the time scale for the transcriptional response in the range of one hour, similar to the expected time scale of mRNA half-lives. This should be mentioned.

Figure 4B. Please provide a color legend.

Figure EV2. Indicate the units of k_{eg} . presumably it is 1/hour but the axis in panel A is in minutes.

Since it is usually not possible to resolve k_t and k_i from smFISH distributions (as mentioned in the Methods), it would be easier and possibly more accurate to use the burst size as one of the free parameters and use the approximation of short bursts. Especially since in the end the authors discuss mainly the bursts and not k_t or k_i .

"We found no significant reductions in noise after normalizing for cell area". This is surprising since by default (unless there is an active buffering mechanism) the coefficients of variation squared add up. What is the CV of the cell sizes? Please add this to the text.

Point-by-point Reviewer Responses**Reviewer #1:**

This is an interesting manuscript that studies NFkB target gene expression and its relation to underlying noise in T cells (Jurkat cell line). The authors chose an important pathway and cell type, relevant to medical problems and immunology, and studied single cell and population based gene expression variability and supplemented their experiments with modeling. The main conclusion of the paper is that TNF stimulation induces NFkB gene expression by increasing promoter burst size and not burst frequency. This is an interesting finding and is well supported by the analyses in the paper.

We thank the reviewer for the positive comments regarding our main conclusion.

Their second surprising conclusion is that TNF induced positive feedback can result in increasing cell to cell variability in gene expression. While this is suggested by their theoretical analyses, it is not supported by experiments. The TNF transcripts produced by the cells are very low (10-15 per cell) and it is not clear if this can lead to sufficient TNF production-release-and

feedback by restimulating the cells. This should be explored experimentally before accepting this paper for publication. They can measure TNF production by the same cells upon TNF stimulation, for example by single cell ELISA (ELISPOT).

We acknowledge that our model was too speculative without experimental validation. To collect experimental evidence of the impact of TNF positive feedback, we blocked secretion with brefeldin A (BFA) and then remeasured *Tnf* transcription by single molecule RNA FISH (smFISH) following TNF treatment. We found that BFA reduced *Tnf* mean transcription and the inferred burst size at 2 hours as would be expected if a secreted signal were positively regulating transcription (new Fig. EV5A). We also measured TNF protein by intracellular cytokine staining and flow cytometry. When we treated cells with TNF for 8 hours while simultaneously blocking secretion with BFA, we found that a small fraction of cells increased intracellular TNF. Importantly, when we treated cells with TNF but delayed adding BFA for 4 hours so that extracellular signaling could proceed, this fraction of TNF+ cells significantly increased (new Fig. EV5B). Together these experimental results support a role for positive feedback (page 14, paragraph 2).

Other main conclusion of the paper is that gene expression noise (variability) is inversely proportional to mean copy number of the transcripts expressed by the gene (first two figures), but this is not surprising. This is a simple consequence of poisson-like distributions, and was observed before.

We thank the reviewer for pointing out that it would be useful to directly compare our observations to the expected behavior of a Poisson distribution. While both the empirical relationship we observe and the Poisson distribution display an inverse correlation between noise and mean in the basal state, noise decreases less with mean than what would be expected for a Poisson distribution (Poisson trendline now added for reference in Fig. 11, Fig. 2D-E, and Fig. EV2A). Importantly, TNF stimulation either maintains this non-Poissonian relationship (Fig. 2E) or causes it to deviate further (Fig. 2D). This second observation in particular was unexpected, and led us to the main insight of the paper—that TNF activation of NF- κ B primarily affects burst size. We revised the text to highlight how our observations deviate from Poisson behavior (red text, p. 6, 8).

In conclusion, this paper would be a good candidate for publication in MSB if the authors can test that their positive feedback idea is supported by experiments (or not). If the experiments fail to show such strong TNF production, they can either caveat or remove the last portion of the findings which seem to be additional to the main finding anyway.

Reviewer #2:

This manuscript by Bass and colleagues describes the transcriptional bursting response of NF- κ B target genes by using RNA FISH after TNF α stimulation. They conclude that immediate early response genes modulate their burst sizes in response to stimulation, in contrast to later responding genes that modulate their burst frequency. The authors link the increased burst sizes to Pol II pausing. The findings are interesting and novel, and the data is generally of good quality.

We thank the reviewer for these positive comments regarding our manuscript.

However, data presentation and discussion are often confusing (see comments below), and the changes in pol II pausing as main cause of changes in burst size is not entirely convincing. The authors should address the points below to make their manuscript suitable for publication.

Major comments:

1. The authors mention that cell size does not contribute to the variability in mRNA expression levels. They then conclude that "shared sources of cellular variation are less important than gene-specific noise sources". This is an overstatement, since there are many other sources of cellular variation. For example, they ignore cell cycle progression, which is typically a very strong contributor to variability in gene expression. This should be addressed in their manuscript.

We thank the reviewer for pointing out that our conclusion regarding shared sources of cellular variation was an overstatement based on the data presented. To account for additional sources of noise, we analyzed nuclear size as a proxy for cell cycle (Chu et al, 2017; Padovan-Merhar et al, 2015)) and we calculated nuclear vs. cytoplasmic mRNA Fano factor to explore the effect of nuclear export (Hansen et al, 2018). We observed no significant change in heterogeneity as measured by CV when normalizing to cell or nuclear area (Fig. EV2C) and a small attenuation of noise in the cytoplasm compared to the nucleus (Fig. EV2B). Finally, we measured *Nfkb1a* and *Tnf* expression in the same cells with smFISH at 0, 1, and 2 hours of TNF stimulation. We found that correlations between these two targets were significant, albeit low ($r = 0.34$ in the basal state and is reduced following TNF stimulation, $p < 0.001$). This is consistent with a shared upstream signaling pathway and transcription factor NF- κ B, but also a significant contribution of target-specific noise. Based on these additional analyses, we conclude that, while there is some impact of shared sources of cellular variation, they appear to contribute less to overall transcriptional noise than gene-specific sources (page 7, paragraph 2).

2. The claims around the effects of A485 and JQ1 are confusing.

- "These changes were more pronounced for Tnf, for which A-485 pretreatment completely eliminated the long-tailed distribution of cells expressing high numbers of mRNA, consistent with its impact on Tnf Fano factor". I don't understand how the authors come to this conclusions when looking at Fig.5E and 5F. To me there is no visible difference between A485 and JQ1 on the figure.

We agree that the effect on the tail of the distribution is similar for A485 and JQ1. In this revision, we emphasize how A-485 raises the fraction of cells that express no *Tnf* transcripts in response to TNF from 2% to 10%, while JQ1 does not affect this metric (Fig. EV4C). This suggests that the drugs are affecting different molecular steps in transcription, which we explain more in response to the next point (page 13, paragraph 3).

- Fig.5C: changes in RNAPII Ser5P are rather small and do not scale with changes in burst sizes for the two genes. The authors should explain what they expect when they use JQ1 in terms of amount of paused PolII and how this relates to burst sizes

We thank the reviewer for pointing out that our expectations were not clearly explained. Previous work showed that burst initiation precedes polymerase recruitment (Bartman et al, 2019). Because A-485 inhibits HAT activity, pretreatment with A-485 prior to TNF stimulation lowers H3 acetylation. Thus, we expected that TNF treatment following A-485 pretreatment would require an increase in burst initiation prior to polymerase recruitment. In other words, we expected to observe an increase in TNF-mediated burst frequency and a decrease in burst size following pretreatment, which is in fact what we observed.

In contrast, the BET inhibitor JQ1 blocks recruitment of the transcriptional activator proteins BRD2, BRD3, and BRD4. Previous work showed that it can inhibit multiple facets of gene regulation, including polymerase pause release and enhancer activity (Shi and Vakoc, 2014; Belkina and Denis, 2012; Stonestrom et al., 2016). In the same study cited above (Bartman et al, 2019), bursting was analyzed following treatment with JQ1 and it was found to decrease both the rate of burst initiation and polymerase pause release, but it did not appear to change the rate of RNAPII recruitment. Thus, we expected to observe a reduction in burst frequency and also burst size. However, when we treated cells with JQ1 in combination with TNF, we found that burst frequency increased, contrary to expectations. Our data appear to confirm the multifactorial activity of JQ1, but are hard to interpret biologically (pages 12-13).

- Fig.5G:

a) *"For Tnf, A485 pretreatment prior to TNF stimulation caused an increase in burst frequency without any change in burst size compared to the basal state (Fig. 5G, H)". I don't see the data of pretreatment prior to TNF stimulation anywhere in the figure, so I am confused.*

We apologize that our results were unclear. We only present smFISH measurements for 1 hour of TNF stimulation *with* A-485 pretreatment (TNF+A485) and without (TNF only). We do not present smFISH measurements for A-485 pretreatment only. We have clarified the text accordingly (pages 11-12).

b) *Which panel corresponds to which gene ? It seems that A485 decreases burst sizes even more than JQ1, but the authors focus their conclusions on pausing in relationship with burst sizes.*

We apologize that the labels were missing from Fig. 5G. As described above, because histone 3 acetylation and chromatin opening (i.e, burst initiation) occur before RNAPII recruitment, we assume that HAT inhibition with A-485 will affect both. Indeed, we can see that an increased number of cells no longer produce any transcript. This is specific to A-485 pretreatment and it would follow that the burst size would also decrease considerably.

3. *The last part on the positive feedback loop is difficult to follow. The authors should improve the way they explain how their simulations in Fig.6B-E fit with the real data*

In response to comments from multiple reviewers, we collected additional experimental data on intracellular TNF protein levels and how *Tnf* transcription and TNF protein are reduced when extracellular signaling is blocked with brefeldin A (BFA). Please see our response to Reviewer 1's first point for a description of the data collected. These data allowed us to adjust our model fit of *Tnf* transcription to reflect the burst size reduction observed at 2 hours without positive feedback (i.e., in the presence of BFA) and then tune the strength of the positive feedback to

match mean transcript measurements with positive feedback contributing. With this model, we could explore how cell-to-cell heterogeneity of protein expression would vary with and without feedback. Our results qualitatively match our protein measurements by flow cytometry. We have substantially revised this section of the Results and Methods to reflect these changes (page 14, paragraph 2).

Other comments:

- *The authors mention chromatin accessibility while looking at histone acetylation. While this correlation might generally hold true, they should refrain from mentioning something they do not measure directly (chromatin accessibility), or even better, perform ATAC-seq to justify their claims.*

We have revised the text to be specifically refer to histone acetylation rather than chromatin accessibility.

- *Fig.1J and 2D: The data presentation should be changed, it is very difficult to understand which datapoint corresponds to what.*

We have revised these graphs so that all data points can be identified.

- *The author state at the bottom of page 5: "Average mRNA levels were inversely correlated with AcH3:H3 ratio.". This is confusing, I don't see this inverse correlation anywhere, and it seems to me that it should be the opposite.*

Thank you for catching this mistake. We have corrected the text.

- *Fig.2A: The duration of the TNF treatment is not specified*

We have added the time point (1 hour) to the figure legend.

- *Page 9, top: they cite Fig.EV4 for distribution plots - do they mean Fig.EV3 ? Also they should show TNFa stimulated distributions for Nfkbia mRNAs.*

We have corrected the text to refer to the appropriate figure (Fig. EV3). We have also added the TNF stimulated *Nfkbia* distributions to this figure.

- *Fig.4: A: which are the dashed ones? B: what does the color code mean?*

Based on comments from multiple reviewers, we changed the display of our ChIP data to bar graphs so that it is easier to see the differences between targets and we can mark points that are significantly different from baseline (revised Fig. 4).

- *Page 11: "a large reduction in the number of cells expressing much higher than the mean for both genes (Fig. 5D)". Do they mean Fig.5E?*

We have rewritten this section and corrected the previous mistakes.

Reviewer #3:

The paper by Bass et al analyses the effect of TNF stimulation on the variability of the NF-κB-target genes. The authors demonstrate that TNF affects the chromatin and polymerase interactions at the specific gene promoter, which ultimately leads to changes in the respective mRNA distributions (and variability). Authors then use telegraph model to understand the modulation of transcriptional bursting in terms of burst size and frequency (in particular in resting cells vs cells stimulated with TNF). They demonstrate that TNF stimulation affects burst size as well as burst frequency in different gene subsets. Finally, they use a model of positive TNF feedback to evaluate the role of transcriptional bursting in the generation of cellular heterogeneity.

Overall, this is a timely subject and elements of the manuscript are performed to a high standard. I do appreciate the relatively large ChIP dataset and use of inhibitors.

We thank the reviewer for these positive comments.

However, other elements in my opinion require more work. I have questions regarding the justification and application of the telegraph model and fitting protocols. I also believe that the paracrine feedback analysis is potentially interesting but should be validated by additional experimentation. Moreover, I do struggle to access how robust are specific mRNA datasets, from reading the manuscript it is unclear whether data is replicated, how many cells are available, and how consistent are replicates (in terms of mean, CV, Fano factor and other estimates).

Specific comments:

Fig. 1.

Not clear from the text if the smFISH data is replicated, CIs are produced by bootstrapping. In G some of the CIs are missing. Are statistics calculated based on individual replicates show similar differences as those obtained from bootstrapping.

We thank the reviewer for pointing out that we need more transparency in reporting our experimental data. We have added a table clarifying the number of replicates and total number of cells analyzed for each target and condition (see Table 1 in Methods, page 25). We have also added this information into the legends each time a data set is introduced. Because some of the targets and conditions were only measured once, we chose to bootstrap 95% confidence intervals for all samples. To determine how well bootstrapped CIs reflected error from replicate samples, we compared bootstrapped CIs to the standard deviations we obtained from multiple independent experiments and we found similar results (Appendix Figure 3).

"The slope of this conserved mean-versus-noise trendline suggests non-Poissonian stochastic transcription rather than continuous transcription in inducible NF-κB targets (Dar et al, 2016; Singh et al, 2010). " For clarity, can the Poisson model be plotted (i.e. $CV^2 \sim 1/\mu$) on the graph. This will highlight how much more noise there is in the data.

Thank you for this suggestion, which was also raised by Reviewer 1. We have added the Poisson trendline to the plots in Fig. 1-2 and we have included mean-variance plots in Fig. EV2. Please see the above response to Reviewer 1 for a discussion of the differences.

Fig 2.

*"We found that for the NF- κ B targets that increased mean without a significant reduction in noise, we observed that in some cases they moved outside the basal trendline, especially at 2 hours (Fig. 2D, left). In contrast, *Il6* and *Csf2* remained within the trendline of the basal measurements upon TNF treatment for 2 and 4 hours as noise decreased with an increased mean (Fig. 2D, right). Overall, this suggests that NF- κ B differentially regulates transcriptional noise at different target genes following TNF stimulation. "*

These panels are difficult to understand; can the authors highlight different genes, what are the specific conditions that do not follow the trend.

We have updated this plot so that all of the conditions can be identified.

The suggestion that heterogeneity may be altered somehow is rather vague. Recently, (Bagnall et al., 2020), demonstrated that NF-kappaB dependent genes in response to stimulation exhibit conserved linear mean-variance relationships, which describe how noise changes in response to stimulation/perturbation. The genes considered here should follow similar constraints (given linear mean-variance, there should be a nonlinear CV²-mean relationship for each gene), which can aid with interpretation of the data.

There is indeed a conserved mean-variance relationship that deviates from Poisson (now presented in Fig. EV2A, left). Unlike Bagnall et al, 2020, we find that all of our target genes (unstimulated and stimulated) largely fall along this trendline (i.e., we do not see distinct trendlines for different genes). This might be due to lower levels of gene expression and fewer signaling pathways/TFs activated in response to TNF versus Lipid A (as used in that study). If we plot CV² versus mean on a linear scale, we indeed see a non-linear relationship, in which different burst sizes would be expected to occupy different nonlinear curves (now presented in Fig. EV2A, right). Plotting these on a log-log graph linearizes these curves and increases in burst size can be visualized as movement to the right of the trendline (as in Fig. 2D). Given the low levels of gene expression in our system, we found that the log-log graph makes it easier to visualize the basal trendline and also emphasizes how TNF affects mRNA distributions in relation to this trendline. The observation that TNF moves different genes in distinct directions in relation to this trendline indicates that TNF is differentially affecting heterogeneity, and we add detail to this observation with each subsequent figure. The value of reporting the trendline analysis is similar to that proposed in Bagnall et al, 2020: we see evidence for constraints in these conserved trendlines. In our case we see differences in sets of genes (*Il6* and *Csf2* vs. *Tnf*, *Tnfaip3*, etc) while Bagnall et al observed differences in *Tnf* vs. *Il1b*. Although there is not yet enough overlapping data and conditions to determine if these trends would converge, we think it is likely that they would. We have added a paragraph in the Discussion to draw out some of these comparisons (page 17, paragraph 3).

"We found no significant reductions in noise after normalizing for cell area (Fig. EV1C), in contrast to other targets for which single-cell mRNA expression has been shown to correlate with cell size (Bagnall et al, 2018; Padovan-Merhar et al, 2015). This lack of correlation with cell

size suggests that, for these inflammatory gene targets, shared sources of cellular variation are less important than gene-specific noise sources. " The statement somehow lacks precision, the cited paper looked at TNF and NFKBIA, which are the same genes.

We agree the statement was too strongly worded for what we reported. In this revision, we analyzed additional shared sources of variation, including nuclear size as a proxy for cell cycle, nuclear vs. cytoplasmic mRNA to explore the effect of nuclear export, and upstream NF- κ B signaling by measuring correlations between target genes (revised Fig. EV1). Please see our response to Reviewer 2 for a more complete description.

Similar to Bagnall *et al*, 2018, we measured *Nfkiba* and *Tnf* in the same cells using multiplexed smFISH. We also found a significant correlation between these two targets, but the r value was much lower than observed in Bagnall *et al* for the same targets following LPS stimulation in macrophages ($r = 0.34$ in the basal state and is reduced following TNF treatment; Fig. EV1D). This is likely attributable to differences in cell type and stimulus, as Jurkat T cells exhibit ~10-fold lower gene expression in response to TNF as compared to macrophages in response to LPS, and intrinsic noise sources might be more dominant at these low expression levels.

Fig. 3:

*There is no justification for using an intrinsic noise model here. Authors should be able to look at their smFISH data, quantify the level of intrinsic vs extrinsic noise, and determine whether the telegraph model is appropriate in this case. This is important because recent evidence suggests a substantial contribution of the extrinsic noise in the NF- κ B-dependent transcription (Zambrano *et al*, 2020), i.e. transcription kinetics appears to be pre-determined in a subset of cells.*

The mean-variance trend that we see is well within the expected relationship for bursty gene expression (Skupsky *et al*, 2010; Singh *et al*, 2010; Dar *et al*, 2016), and we agree with the reviewer that it is important to show this in order to justify the use of the random telegraph model, which we have now done in this revision (Fig. EV2A). Based on the recent work of Zambrano *et al*, and even previous papers from our lab (Wong *et al*, 2018 and Wong *et al*, 2019), we agree that variations in NF- κ B signaling are correlated to transcriptional output, although which aspects of the dynamic signal are most predictive of the dynamic response remains a point of some debate. However, influence of the upstream signal does not rule out a difference in transcriptional distribution. In fact, we showed that NF- κ B translocation was similarly predictive of *Tnfaip3* and *Il6* transcription in Jurkat cells after TNF stimulation (Wong *et al*, 2019), despite the fact that these targets have very different distributions as we report here. In other words, the differences in noise (Fano factor, CV, etc) exist in our data despite a shared upstream signal and before we invoke an intrinsic noise model. Although it ignores these shared signaling inputs, the random telegraph model provides a convenient way to interpret our results. Given the low correlation we measure between *Nfkbia* and *Tnf* that is reduced after TNF treatment (Fig. EV1D), we think the use of this model is justified.

*"Using a previously described method (Rat *et al.*, 2006. Skupsky *et al.*, 2010), we held the transcription rate kt constant across all conditions and reported b . "*

There are more accurate methods that allow fitting all the parameters in question, especially those taking into the account temporal regulation and measured basal distributions- see for

example (Gomez-Schiavon et al., 2017). The assumption that the k_t does not change, in particular between basal and induced conditions seems very strong. Authors imply that a basal state works in a different mode than the inducible state, therefore why k_t should be the same. For example, the TNF mRNA fitting by (Bagnall et al., 2020) show in fact large changes in the k_t value between basal and stimulation conditions.

We use MLE fitting of the probability density function of the random telegraph model to infer burst size, which is equivalent to k_t/k_i in the model, and burst frequency, which is equivalent to k_a . We do not know the relative contribution of changes in k_t vs. k_i to burst size, and we do not claim that k_t is remaining constant. Rather, we hold k_t constant in order to simplify the MLE fitting (i.e., we can fix k_t and k_{deg} and fit k_a and k_i). We have clarified this in the revised text.

The only conclusion we make from this analysis regards the relative contribution of burst size vs. burst frequency to the overall increase in transcription observed following TNF stimulation. To emphasize this point, in this revision we now also compute burst size and burst frequency based on moments, as was done in Bagnall et al, 2020, and which does not require fitting any model (Fig. EV2B and C, discussed on page 8 paragraph 3). This analysis reveals the same trends as our MLE fitting. Because we are not making additional claims from this model beyond the burst size/burst frequency comparison, we do not think more sophisticated model fitting methods would add value to our study and might lead us to overinterpret our relatively limited dynamic measurements.

*It is not clear is whether the model assumed two or one alleles, which will affect fitted parameter values (also model with 2 alleles might allow fitting *nfkbia* count data).*

We thank the reviewer for pointing out this omission. We assumed two independent alleles for each gene, and we have added this to the Methods (page 24). We have also revisited the model fits for *Nfkbia* and found conditions that reasonably fit our distributions. It is notable that these inferred burst sizes differ more from those calculated based on the Fano factor than for any of the other targets. We have added the model fits to *Nfkbia* distributions before and after TNF treatment to Fig. EV3.

*Also, the authors use " k_a " as the estimator of the bursting frequency. This is only an approximation and assumes that the gene is in the 'bursty' regime, roughly $k_{off} \gg k_{on}$. Therefore, the authors should check if this assumption is actually satisfied with their data. Otherwise, a more generic estimator for frequency should be used $f = k_a * k_i / (k_a + k_i)$ (Nicolas et al., 2018).*

As described above, we have now included more analysis demonstrating that our data exhibit mean-variance and mean-noise that fall within the 'bursty' regime of transcription. In addition, as described above, we calculated burst sizes and burst frequencies based on the moments of the distributions (Fig. EV2B and C), which produces trends that match what we infer from the random telegraph model.

Fig. 4

Are the changes (basal vs. stimulated) statistically significant?

We have changed the display of the ChIP data to indicate which points are significantly different from basal following TNF stimulation.

Panel B needs an indication of the scale (i.e., blue-red levels). Relationship between CV/Fano factor and burst size/frequency highlighted in the legend should be explained explicitly

Thank you for pointing out the scale bar omission. We have added the color scale bar, removed the reference to CV and Fano factor, and more clearly explained the relationship to burst size and burst frequency.

Fig. 5

"To perturb histone acetylation, we pre-treated Jurkat cells with the histone acetyltransferase (HAT) inhibitor A-485, a specific inhibitor of the HATs p300/CBP that are recruited by NF- κ B (Fig. 5A) (Gerritsen et al, 1997; Lasko et al, 2017). We found that treatment with A-485 for 4 hours decreased Ach3 levels at the Tnfaip3 and Tnf promoters but did not significantly affect total H3 levels as measured by ChIP-qPCR (Fig. 5B)." Statistical analysis is performed (also in C), a pairwise t-test based on 2 replicates (as legend says). I would not deem this as an appropriate use of statistics.

We have updated these subpanels to show the individual replicates as well as mean and standard deviation. We no longer report p-values on those with n = 2 replicates.

Panel D- no indication whether the data was replicated, how many cells are available in this dataset, are means of replicates per condition consistent?

We have added a Table that reports all of the sample sizes and replicates for the smFISH data (page 25). We have also added this information into the legend each time a new data set is introduced.

Authors state: "Overall, these inhibitors decrease TNF-stimulated transcription while differentially affecting mRNA distributions. " In fact, the distributions in E corresponding to both inhibitors appear to be quite similar to each other, as indicated by fitted bursting characteristics in F and EV4.

We have substantially revised this section of the results to more clearly describe our expectations for how the drugs would affect transcriptional bursting, and how our observations were (or were not) consistent with these expectations (pages 11-13).

Panel G- labelling is missing, not sure were is TNF vs TNFAIP3. Authors state "Fitting mRNA distributions for TNF treatment following pre-treatment with A-485 to the theoretical pdf of the two-state model, we found that, in response to TNF treatment, A-485 decreased burst size for Tnfaip3 while not affecting burst frequency. For Tnf, A485 pre-treatment prior to TNF stimulation caused an increase in burst frequency without any change in burst size compared to the basal state (Fig. 5G, H)." In fact panel B suggests that the burst size is decreased.

We apologize for the myriad mistakes in this figure in our initial submission. The figure has been substantially reworked and these errors have been corrected.

The authors try to draw generic conclusions based on 2 inhibitors and behaviour of two genes. I find it quite difficult to follow and accept, especially not knowing how robust this dataset is. A number of points is often made based on a correlation between a panel of genes and CHIP data.

We have revised this section. While we think that our conclusions based on inhibition of HAT activity are robust, especially when combined with our previous results (Wong et al, 2018), we agree that the JQ1 data is inconclusive and would require more specific perturbations to explore further. Therefore, we have backed off this claim in the paper and removed it as a specific point in the abstract.

Fig. 6.

As TNF can stimulate its own transcription, we included positive feedback from the newly produced TNF on the cell that produced it (simulating autocrine signaling) to explore its effects on cell-to-cell heterogeneity (Fig. 6A). We modelled the addition of exogenous TNF as a time-dependent change in k_t , the mRNA production rate, in order to match the change in burst size inferred from our smFISH distributions. "

Here the authors provide a theoretical study of the potential effect of the positive TNF feedback on the transcriptional bursting. I would expect that positive-feedback should modulate responses by contributing to the ongoing NF-kappaB response/variability, which then modulates transcriptional bursting by affecting k_{on} , k_{off} , and k_t rates (directly through the probability of NF-kB binding and dissociation or indirectly by further changing chromatic state, as authors suggest). Therefore, the explicit assumption about the effect of feedback on the k_t rate seems very strong. Especially that in Fig. 3 authors assume that this rate does not change between basal vs TNF stimulated conditions.

We agree with the reviewer that TNF positive feedback would affect many aspects of NF- κ B, but we think that modeling these is beyond the scope of the point we wanted to make for this study. Rather, we sought to use a very simplified model of positive autoregulation to explore the idea that positive feedback might differentially affect distributions with different shapes. As noted above, we do not assume that k_t is constant following TNF stimulation, but rather we conclude that the burst size (k_t/k_i) changes significantly upon TNF stimulation. In Fig. 6, we **do** make the simplifying assumption that the burst size increase is due solely to increased k_t in order to model TNF stimulation and feedback.

More importantly, the validation of these simulations is important. Does Jurkats produce a meaningful amount of TNF (in relation to the 20ng/ml stimulation dose) in the time scale of the experiment? Is there a difference in the mRNA distribution when the feedback (TNF or generic) is blocked experimentally, does it induce predicted changes to transcriptional bursting? I feel that without this validation the modeling is very speculative. Ultimately the modeling demonstrates that if we change burst size or frequency, we do affect the heterogeneity. But this is a property of the telegraph model. (Bagnall et al., 2020) demonstrates that the modulation to

burst size and frequency follows relationships can be predicted by the gene-specific linear relationships. This might help to interpret the data presented in this manuscript.

We have now added experimental evidence for a role of extracellular signaling in amplifying mRNA and TNF protein in Jurkat cells following TNF stimulation (Fig. EV5A-B). Please see the response to Reviewer 1's comment above for a detailed description of the data collected.

As the reviewer points out, the random telegraph model predicts that activating transcription by changing burst size vs burst frequency will result in different mean-noise relationships (Fano increases with burst size, while CV decreases with burst frequency). The purpose of the model in Fig. 6 is to first show how positive feedback might further amplify and exacerbate these differences. This is motivated by our previous work in HIV latency, in which we showed that viral-mediated positive feedback amplified and activated viruses that exhibited a burst size increase following TNF stimulation (as measured in the absence of feedback), but did not efficiently amplify viruses that exhibited a burst frequency increase (Wong et al, 2018). Although TNF production is low in Jurkat cells, we do measure a role for positive feedback via extracellular signaling, and our simulations show that this feedback further skews the distribution of protein levels, and predicts a subset of highly functional cells (i.e., relatively high TNF producers as compared to the mean population).

Bagnall, J., Rowe, W., Alachkar, N., Roberts, J., England, H., Clark, C., Platt, M., Jackson, D.A., Muldoon, M., and Paszek, P. (2020). Gene-Specific Linear Trends Constrain Transcriptional Variability of the Toll-like Receptor Signaling. *Cell Syst* 11, 300-314 e308.

Gomez-Schiavon, M., Chen, L.F., West, A.E., and Buchler, N.E. (2017). BayFish: Bayesian inference of transcription dynamics from population snapshots of single-molecule RNA FISH in single cells. *Genome Biol* 18, 164.

Nicolas, D., Zoller, B., Suter, D.M., and Naef, F. (2018). Modulation of transcriptional burst frequency by histone acetylation. *Proc Natl Acad Sci U S A* 115, 7153-7158.

Reviewer #4:

Report for "TNF stimulation primarily modulates transcriptional burst size of NF- κ B-regulated genes"

The authors perform an analysis of the NF κ B transcriptional response following TNF stimulation in Jurkat cells. NF κ B is obviously an important transcription factor mediating inflammatory and stress responses. While the transcriptional response parameters and bursting properties of the HIV long terminal repeat (LTR) promoter have been fairly extensively studied, less is known about the responses of endogenous NF- κ B targets. It is well known that the analysis of transcriptional responses in mammalian cells at steady-state boils down to the description of the burst frequency and size, which can be done using live reporters (for example MS2) but also with smFISH even though the latter is less direct. Here the authors study transcriptional bursting at six endogenous NF- κ B target promoters before and after TNF stimulation, including both primary inflammatory genes (direct targets) as well as two secondary genes. Upon TNF stimulation increases, mean transcription of the direct targets increased while noise (CV) did not significantly increase, which is equivalent to the statement that stimulation primarily increases burst size while maintaining burst frequency. It is then argued that the increase in transcriptional

burst size is linked to TNF-dependent regulation of RNAPII pausing. While the notion that activation of certain signaling pathways transiently increases burst sizes has been described before, the specific results on NFkB transcriptional response are interesting and timely.

We thank the reviewer for the overall positive assessment of our results.

There are methodological and presentation issues that need to be addressed, which will make the manuscript significantly stronger.

*** Major ***

1) *The smFISH images and associated quantifications in 1F,E and 2A-C are quite atypical and seem of heterogenous quality. In Figure 1 it appears that the FISH signal is essentially in the nucleus and it is unclear what a single mRNA spot looks like. Why is the background signal in the FISH channel so high, particularly in the nucleus? Is there a potential issue with the hybridization conditions or probe specificity, or is this reflecting the image filtering?*

We thank the reviewer for pointing out that our images in Fig. 1-2 are confusing given the quantification of our data. We should have more clearly explained that our study combines images that were taken with a spinning disk confocal microscope (one image set for each target and the only image set for *Il8* and *Csf2*) and images taken with a widefield microscope (all replicate images for the other targets). Additionally, the *Il8* and *Csf2* probe sets were labeled with a different fluorophore. *Il8* and *Csf2* were labeled with fluorescein, which has a higher background than Quasar 670, which was used for to label the other probe sets. Although using a confocal microscope somewhat reduces the background of fluorescein, it is not complete, as is evident in the images. Despite these differences, we are confident that our analysis is able to differentiate between punctate smFISH spots and more diffuse background fluorescence, because we carefully set our intensity thresholds by comparison to unstained control cells. We have added these details to the Methods sections on smFISH (pages 21-23). We have also added a sentence to the legend that indicates that *Il8* and *Csf2* were imaged with a different probe/microscope combination to explain why these images appear different.

Concerning the quantifications, comparing Nfkb α Il8 and Csf2, it is not clear why these genes differ quite strongly in the number of counts per cell. In general the correspondence between the images and quantifications is difficult to grasp.

Similarly the comparison between Figure 1E and 2A is confusing. In 2A (Tnfaip3), mRNA dots are clearly visible while this was not the case in 1E. Il8, which clearly has the strongest signal in 2A, is reported with only max 10 mRNA counts.

As explained above, the reason that the images for *Il8* and *Csf2* do not appear to correspond by eye to their quantification is because these were labeled with a fluorophore that has higher background than the fluorophore used to label the other genes. These different image sets are analyzed with different intensity thresholds to quantify spots, and these are set by testing different thresholds on multiple cells and images, including unstained control cells, which differ between the fluorescent channels used. We are confident about our smFISH quantification for *Il8* and *Csf2* because we have previously successfully used confocal imaging of fluorescein-labeled probes to quantify HIV transcription in similarly low transcript settings (Wong et al,

2018). Also, our smFISH quantification agrees with our population level RT-PCR measurements (Fig. 1C). Specifically, *Nfkb1a* and *Tnfaip3* have the highest expression, *Tnf* and *Il8* have similar expression that is lower, and *Il6* and *Csf2* have similar expression that is not detected by RT-qPCR.

Thus, the authors need to significantly improve their presentation and potentially analysis of the smFISH data. In particular more justification and supplementary material is needed, showing more and larger raw (unfiltered) images, and show how the quantification and counting of the mRNA in each cell was done. It is not enough to simply refer to the FISH-quant software.

In response to these concerns, we have added a new Appendix figure that displays full size images before and after TNF stimulation for all of our target genes before and after the image processing (Appendix Fig. S1). In reference to this figure, we have also added more explanation in Methods about how images are processed and intensity thresholds are set (pages 22-23). The Dual Gaussian filtering method first uses a large Kernel (5 pixels) to blur the image for background subtraction, then uses a small Kernel (0.5 pixels) to enhance small features such as smFISH spots in the background subtracted image. Filtered images are then analyzed with a range of threshold intensity values that are tested on multiple images, including unstained controls, to identify spots using a local maximum calculation during pre-detection using the FISH-Quant GUI. The intensity threshold chosen for each set of images varies with the microscope, fluorophore, specific probe set, and experimental condition (Mueller et al, 2013; Tsanov et al, 2016).

2) Besides a general statement in the Methods (see next paragraph) it was hard to find the information on how the authors handled replicates of smFISH experiments for each Figure. There's a short statement in the Figure EV1 caption, but it should be included systematically. It should also be stated clearly what variability the bootstrap confidence intervals in 1G-I, 2B 5G-H are capturing. It seems that the data are mostly analyzed in a pooled manner, except in Figure EV1C. In the methods it is stated that "All smFISH experiments included a sufficient number of cells to characterize the transcript distributions ($n > 100$ cells) and results were confirmed with independent biological replicates." but it was difficult to understand what is meant by 'confirmed' for each experiment shown.

The number of smFISH replicates varied by condition and target. To make our data more transparent, we have added a table in Methods that reports the number of replicate experiments and total number of cells analyzed for each gene and condition (Table 1, page 25)). Because a few conditions were measured only once by smFISH, we chose to use bootstrapped 95% confidence intervals for all samples. To assess how well bootstrapped CIs reflect experimental variability, we compared the error associated with mean expression for *Nfkb1a*, *Tnfaip3*, *Tnf*, and *Il6* when either pooling three replicate experiments and bootstrapping 95% confidence intervals or averaging the means calculated separately from each replicate (see new Appendix Fig. 2). We found that these were very similar.

3) Panel J: "The slope of this conserved mean-versus-noise trendline suggests non-Poissonian stochastic transcription rather than continuous transcription". The interpretation of the CV - mean relationship could be strengthened. The fact that the relation is linear in log-log (or power law in the original variables) is interesting, but it's really the values of the slope and intercept

that matter. Poisson noise corresponds to a slope of -1 and zero intercept. Therefore this results need to be interpreted with respect to the slope and intercept of the fitted line. For example, for the telegraph model the CV^2 equals $(b+1)/m$, where m is the mean and b the burst size. This is partially discussed in the section on burst sizes but could be started here.

Thank you for this suggestion, which was also raised by Reviewer 1. We have added the Poisson trendline to the plot. Please see the above response to Reviewer 1 for a discussion of the differences.

4) There is also a statement on Figure 2D in the section on burst size "Interpreted in this way, the mean-versus- noise plots suggest TNF differentially regulates burst frequency and burst size across different NF- κ B target genes (Fig. 2D) (Sanchez and Golding, 2013)." This is interesting but is hard to understand without further explanation.

We have computed burst size and burst frequency based on moments, which relies on distribution statistics and assumes no underlying bursting model, to further support the conclusion that different genes exhibit different bursting behaviors (Fig. EV2B and C) (Bagnall et al, 2020).

5) When introducing the random telegraph model as a way to analyze the noise, the authors should clearly state that this does not take into account extrinsic noise or other form of heterogeneity. This is a very strong assumption and that should be justified (cell cycle states, microenvironment, etc). Departure of the data from this assumption can easily lead to wrong conclusions after fitting. How do the authors justify this approximation, in particular since it appears that there is heterogeneity in the proportion of responders? In principle, more complex models would be needed to take into account such heterogeneity.

To more thoroughly assess extrinsic sources of noise in addition to cell size, we added analyses of how noise varies with nuclear size (as a proxy for cell cycle), with nuclear export, and with NF- κ B signaling as measured by correlations between target genes (Fig. EV1). Overall, we did not see a strong role for extrinsic noise (please see comments for Reviewer #2 for more discussion of this point). Overall, we conclude that gene-specific sources of noise contribute more than extrinsic cellular noise.

Regarding the use of the random telegraph model, we show that the mean-variance trend that we see is well within the expected relationship for bursty gene expression (Skupsky et al, 2010; Singh et al, 2010; Dar et al, 2016), (Fig. EV2A). Although there is heterogeneity in the proportion of responding cells, even our targets with very low expression (Csf2 and Il6) have 80-90% of all cells responding by 4 hours. The outputs from our analysis using the random telegraph model largely agree with the bursting based on moments (new Fig. EV2), lending confidence that the approximations of the model are not significantly altering our interpretation of the underlying biology. Finally, we added a measure of correlation between *Nfkb1a* and *Tnf* to account for shared upstream signaling and we found that the correlation is only moderate in the basal state and is reduced after TNF treatment ($r = 0.34$ and lower after stim; Fig. EV1D). Overall, in this revision, we have included more analysis to justify the use of the random telegraph model. Please see responses to Reviewer #2 and #3 above for additional discussion.

6) Figure 4. The interpretation of an increased level of paused PolIII relies on subtle differences

between s5p and s2p that may not be large enough to make a strong point. It also depends on the location of the PCR primers, which is not discussed in the main text. It would be useful to measure RNAPII in the gene body to confirm that the level of transcription is indeed low in the paused state.

We thank the reviewer for raising this important point regarding interpretation of our ChIP results. We have added the locations of the ChIP primers relative to the transcription start site of the target gene to the table reporting the primer sequences used (Supplemental Table S1). We also added significance values relative to the basal state for all the experimental ChIP data reported in Fig. 4A. The marked and significant increase in RNAPII-S5p for *Il8* and *Tnf*, combined with relatively low expression suggests pausing, but we agree the situation is less clear for *Tnfaip3* and *Nfkb1a*, which are more actively transcribed. While we still think these data suggest an association between RNAPII pausing and burst size, we have significantly reduced the strength of this claim in our revised manuscript. Particularly when combined with the hard-to-interpret results for JQ1 reported in Fig. 5, we think our data motivate additional studies but do not support a strong conclusion at this point. In contrast, the role for basal acetylation of histone 3 is more clearly supported.

7) A recent work by Zambrano et al (iScience 2020, PMID: 33083759) is briefly cited but it would be interesting to more extensively discuss it since this study also performed time courses of endogenous genes by smRNA FISH, as well as live analysis with an MS2 reporter, though in a different cell line. Also, the finding here are reminiscent of other systems where the activation of signaling transiently increases burst sizes, which should be mentioned (Molina et al. PMID: 24297917).

We thank the reviewer for highlighting these relevant papers. Because Zambrano et al 2020 looked at similar targets following TNF stimulation of HeLa cells and Bagnall et al 2020 looked at similar targets following LPS stimulation in macrophages, we added a paragraph in the Discussion to discuss our observations in the context of these papers, both of which found that a more complex model of transcription, including a refractory state, was required to reproduce the trends they observed. We think that this might be due to the larger range of expression in their cell systems (see Discussion, page 17). We added Molina *et al* to our citations of studies reporting a signaling-induced burst size increases in other systems (see Discussion, page 16).

8) A general puzzle is that Figure 1 clearly show that the transcriptional action of the direct targets takes place at very short times (probably <1 hour if one were to look at nascent transcription). Yet, most experiments aimed at elucidating the mechanisms are done at 2 and 4 hours (Figure 5), which thus seems too late.

We originally performed ChIP at 2 and 4 hours after TNF stimulation to compare our results for endogenous genes to our previous work studying the HIV-LTR promoter (Wong *et al*, 2018). We acknowledge that these time points are late relative to transcriptional activation for our 'burst size' target group (*Nfkb1a*, *Tnfaip3*, *Tnf*, and *Il8*). However, because we captured distinct differences between these targets and *Il6* and *Csf2*, we did not go back and collect earlier time points. In this revision, we changed the layout of the data from line vs. time to bar graphs so as not to imply that we have a complete time course.

For the drug perturbation experiments in Fig. 5, we adjusted our ChIP time points to better reflect the timing of transcription. Specifically, we analyzed histone 3 and acetylated histone 3 in the basal state vs. a 4-hour pretreatment with the HAT inhibitor A-485 and we analyzed RNAPII and RNAPII-Ser5P at 1 hour after TNF treatment alone or in combination with the BET inhibitor JQ1.

*** *Minor* ***

"Average mRNA levels were inversely correlated with Ach3:H3 ratio." Isn't Figure 1B showing the opposite? This was confusing as Ach3 is typically a mark that correlates positively with activity.

We thank the reviewer for pointing out this error. We have corrected it.

In Figure 2A and 2D, it is not specified what the time point is.

We have added the time point for Fig. 2A to the figure legend.

Figure 2D. It is not mentioned what the blue and orange dots represent, what are the criteria for coloring dots in blue or orange? Since 6 genes are measured, it was unclear why there are so many dots. The "caption and description in the text needs to be improved.

We have significantly edited Fig. 2D and 2E to more clearly label the genes and time points.

*"The one exception was for *Nfkb1a*, in which the theoretical pdf from the two-state model could not be accurately fit to the TNF-stimulated distributions. This is consistent with the high transcriptional rate induced by TNF that might be outside the limits of the two-state bursting model (Wilson et al, 2017)." This is unclear, there is not restriction on the transcription rate in the two-state model.*

What we meant to communicate is that as transcription rates increase and transcription becomes more continuous, the distribution may no longer be well fit by the bursting model solution. That said, we also revisited the model for *Nfkb1a* and found we were able to reasonably fit our distributions (Fig. EV3). It is notable, however, that these inferred burst sizes differ more from those calculated based on the Fano factor than for any of the other targets.

A potential caveat of the fits is that the theoretical distributions are appropriate in steady-state (or quasi steady-state) situations, which is not the case here as the time scale for the transcriptional response in the range of one hour, similar to the expected time scale of mRNA half-lives. This should be mentioned.

We agree that fitting dynamically changing transcript distributions to a theoretical steady-state distribution is an important caveat to mention and we have added this to the Methods (page 24). We note that we have added analysis of distribution moments and find that they also reveal different trends following TNF treatment (Fig. EV2).

Figure 4B. Please provide a color legend.

Thank you for pointing out this omission. We have added a color legend to Fig. 4B.

Figure EV2. Indicate the units of k_{eg} . Presumably it is 1/hour but the axis in panel A is in minutes.

We thank the reviewer for pointing out the lack of units (which are 1/hour). We have added them to the mRNA decay figure, which is now referenced in Methods as it supports details of how the model was fit (Appendix Fig. 2).

Since it is usually not possible to resolve k_t and k_i from smFISH distributions (as mentioned in the Methods), it would be easier and possibly more accurate to use the burst size as one of the free parameters and use the approximation of short bursts. Especially since in the end the authors discuss mainly the bursts and not k_t or k_i .

We agree with the reviewer that because our main goal was to compare burst size and burst frequency, alternate simpler methods of comparing across distributions would have sufficed. In this revision, we addressed this by comparing burst size and burst frequency calculated from distribution moments to compare trends without first assuming an underlying model structure. We found that these results are similar to the fits from our random telegraph model. We chose to retain these fits as this has been demonstrated as a reasonable estimation method in previous work (Dey Foley et al, MSB 2015) and because we use this two-state model for our simulations in Fig. 6.

"We found no significant reductions in noise after normalizing for cell area". This is surprising since by default (unless there is an active buffering mechanism) the coefficients of variation squared add up. What is the CV of the cell sizes? Please add this to the text.

We have added the CV of cell and nuclear area before and after TNF stimulation to Fig. EV2D.

Thank you for sending us your revised manuscript. We have now heard back from the three reviewers who were asked to evaluate your study. The reviewers are satisfied with the modifications made and they are supportive of publication. Reviewer #3 lists a few remaining concerns and requests for edits, which we would ask you to address in a minor revision.

On a more editorial level we would ask you to address the following.

Reviewer #2:

The authors have significantly improved their manuscript by including new experiments and analysis. In particular, they now show experimental evidence for the positive feedback of TNF in their system. They also show evidence for the limited contribution of some important extrinsic noise sources to the intercellular variability in mRNA counts they observe. In my opinion they satisfactorily clarified most issues that have been raised by all reviewers.

Reviewer #3:

The authors have made substantial improvements to the manuscript. All the main points were addressed.

However, new data/text raise some new questions/comments:

1. Authors now state: "To determine if extracellular signaling amplifies Tnf transcription, we stimulated Jurkat cells with TNF in the presence of brefeldin A (BFA), which inhibits protein transport from the endoplasmic reticulum to the Golgi and thus blocks secretion. We found that BFA modestly reduced transcription at 2 hours following TNF stimulation and also reduced the inferred burst size (Fig. EV5A)."

Appreciate the new data regarding the Tnf positive feedback. The authors demonstrate a change of the mean mRNA and burst size after inhibition of generic secretion (although only 1 replicate is provided for BFA). I presume there is no difference in the bursting frequency after BFA treatment. The authors should be able to demonstrate this based on their data since the change of the burst size is then used as a mechanism in the model. Finally, does the positive feedback model fit measured mRNA distributions from EV5A?

2. " The fraction of responding cells was small, consistent with the low mRNA measurements, but a significant increase in intracellular TNF over control was seen after TNF stimulation (Fig. EV5B)." The data are provided as s.e.m's, which implies that sds are substantially larger. The authors state significant differences in the text, but I cannot find information about a statistical test in this case.

3. Authors state (also in their response to reviewer comments): "Maximum-likelihood estimation (MLE) was used to select burst frequency (k_a) and burst size ($b = kt/ki$) parameters that best fit the measured mRNA distributions to the full analytical solution to the two-state stochastic gene expression model (Peccoud and Ycart, 1995). Although this is a steady-state solution, we use it here to approximate how TNF affects transcriptional bursting (Wong et al., 2018). We assumed two independent alleles for each gene (Raj et al 2006; Suter et al, 2011)." Apologies, but I am confused since the cited work considers 1 allele. If there are two alleles in the model as the authors suggest (Table 2), their Equations 1 and 2 should account for this (but they do not).

Reviewer #4:

The authors provided a complete and detailed revision to the many reviewers' comment they received. The answers to my specific points are satisfactory and I felt that the presentation is much improved and overall much more convincing. I got a bit confused with some of the references to the Figures in the replies, e.g. 1C vs 1B, S2 vs S3, so please check that this is ok in the manuscript. Thank you also for providing new appendix Figures (not listed as expanded view it seems), please make sure these are published.

Point-by-point Reviewer Responses

Reviewer #2:

The authors have significantly improved their manuscript by including new experiments and analysis. In particular, they now show experimental evidence for the positive feedback of TNF in their system. They also show evidence for the limited contribution of some important extrinsic noise sources to the intercellular variability in mRNA counts they observe. In my opinion they satisfactorily clarified most issues that have been raised by all reviewers.

We thank the reviewer for these positive comments.

Reviewer #3:

The authors have made substantial improvements to the manuscript. All the main points were addressed.

However, new data/text raise some new questions/comments:

1. *Authors now state: "To determine if extracellular signaling amplifies Tnf transcription, we stimulated Jurkat cells with TNF in the presence of brefeldin A (BFA), which inhibits protein transport from the endoplasmic reticulum to the Golgi and thus blocks secretion. We found that BFA modestly reduced transcription at 2 hours following TNF stimulation and also reduced the inferred burst size (Fig. EV5A)."*

Appreciate the new data regarding the Tnf positive feedback. The authors demonstrate a change of the mean mRNA and burst size after inhibition of generic secretion (although only 1 replicate is provided for BFA). I presume there is no difference in the bursting frequency after

BFA treatment. The authors should be able to demonstrate this based on their data since the change of the burst size is then used as a mechanism in the model.

We thank the reviewer for a careful consideration of our new data and for pointing out that we need to clarify our assumptions. In the presence of BFA, the burst frequency increases in response to TNF more than what we saw in the absence of BFA. However, the relative change in burst size is still greater. To clarify this point, we added the burst frequency estimates to Fig. EV5A and reference them in the Results. We also added text in Materials and Methods (p. 27) that reads, "We note that [the assumption that burst frequency remains constant] is an approximation that is based on experimental observations that TNF modulates burst size more than burst frequency (Fig. 3C), however it does not fully reflect the data in the presence of BFA (Fig. EV5A)."

Finally, does the positive feedback model fit measured mRNA distributions from EV5A?

To make this easier to assess, we have added violin plots of mRNA distributions at 1 and 2 hours measured experimentally by smFISH and produced via our simulations with and without feedback (new panel Fig. EV5E). Comparing these distributions, one can see that they are qualitatively similar and that TNF positive feedback indeed increases the "tail" of transcription in the simulation (particularly at 2 hours) to more closely resemble experimental measurements.

2. "The fraction of responding cells was small, consistent with the low mRNA measurements, but a significant increase in intracellular TNF over control was seen after TNF stimulation (Fig. EV5B)." The data are provided as s.e.m's, which implies that sds are substantially larger. The authors state significant differences in the text, but I cannot find information about a statistical test in this case.

We thank the reviewer for noting this omission. As stated in the text, the increase in the fraction of cells expressing intracellular TNF in response to TNF stimulation with BFA was significant (8h vs. control, now marked in Fig. EV5B). When BFA was delayed by 4 hours following TNF stimulation, the fraction of cells expressing TNF increased but this increase was not significant due to the large error bars as noted by the reviewer. We added text in the Results to clarify this point: "we saw an increase in % TNF+ cells at 8 hours when BFA was withheld for 4 hours to allow paracrine signaling to occur (Fig. EV5B; although increase not statistically significant). Taken together, [the smFISH and flow data with BFA] support a role for positive feedback in amplifying the response."

3. Authors state (also in their response to reviewer comments): "Maximum-likelihood estimation (MLE) was used to select burst frequency (k_a) and burst size ($b = kt/ki$) parameters that best fit the measured mRNA distributions to the full analytical solution to the two-state stochastic gene expression model (Peccoud and Ycart, 1995). Although this is a steady-state solution, we use it here to approximate how TNF affects transcriptional bursting (Wong et al., 2018). We assumed two independent alleles for each gene (Raj et al 2006; Suter et al, 2011)." Apologies, but I am

confused since the cited work considers 1 allele. If there are two alleles in the model as the authors suggest (Table 2), their Equations 1 and 2 should account for this (but they do not).

This statement was indeed an error and we are grateful to the reviewer for noting it and giving us the opportunity to correct it. While we assumed that the two alleles are independent, we are only modeling one of them. For the two-state promoter model fitting, we assumed bursting was sufficiently infrequent such that bursting events of the two alleles were unlikely to overlap, allowing a reasonable estimate of burst size and an upper bound on the estimate of burst frequency (i.e., up to approximately twice the actual value). This assumption is supported by our empirical observation that we rarely see evidence of two active transcription start sites in our smFISH images, and by our own simulation (now reported in Appendix Fig. S4). We think this is a reasonable approach because we are generally comparing relative changes in burst size vs. burst frequency rather than making claims about the actual values. We further note that when we use moments as an alternative method to estimate burst size and burst frequency, we obtain very similar trends, which further supports our approach.

For the stochastic simulation of *Tnf* transcription in Fig. 6, we continued to model only one promoter activation event, as noted by the reviewer. This approach is consistent with our inferred burst frequency values and—as long as the assumptions outlined above hold—should provide a reasonable approximation of transcription. We also reasoned that if we took the alternate approach and adjusted the model equations to account for two alleles, we would simultaneously adjust the burst frequency by halving it, and so in the end the two approaches are similar. We have added text in the Materials and Methods to clearly state that we are only modeling one allele.

Reviewer #4:

The authors provided a complete and detailed revision to the many reviewers' comment they received. The answers to my specific points are satisfactory and I felt that the presentation is much improved and overall much more convincing. I got a bit confused with some of the references to the Figures in the replies, e.g. 1C vs 1B, S2 vs S3, so please check that this is ok in the manuscript. Thank you also for providing new appendix Figures (not listed as expanded view it seems), please make sure these are published.

We thank the reviewer for these positive comments. The appendix figures will indeed be published in our supplemental materials file.

Thank you again for sending us your revised manuscript. We are now satisfied with the modifications made and I am pleased to inform you that your paper has been accepted for publication.

Corresponding Author Name: Kathryn Miller-Jensen
Journal Submitted to: Molecular Systems Biology
Manuscript Number: MSB-2023-10127